EMBO
Molecular Medicine

# Mutating chikungunya virus non-structural protein produces potent live-attenuated vaccine candidate

Yi-Hao Chan[1,2], Teck-Hui Teo[1,3], Age Utt[4], Jeslin JL Tan[1], Siti Naqiah Amrun[1], Farhana Abu Bakar[1,5], Wearn-Xin Yee[6], Etienne Becht[7], Cheryl Yi-Pin Lee[1,2], Bernett Lee[1], Ravisankar Rajarethinam[8], Evan Newell[7], Andres Merits[4], Guillaume Carissimo[1], Fok-Moon Lum[1] (ID) & Lisa FP Ng[1,9,10,*] (ID)

## Abstract

Currently, there are no commercially available live-attenuated vaccines against chikungunya virus (CHIKV). Here, CHIKVs with mutations in non-structural proteins (nsPs) were investigated for their suitability as attenuated CHIKV vaccines. R532H mutation in nsP1 caused reduced infectivity in mouse tail fibroblasts but an enhanced type-I IFN response compared to WT-CHIKV. Adult mice infected with this nsP-mutant exhibited a mild joint phenotype with low-level viremia that rapidly cleared. Mechanistically, ingenuity pathway analyses revealed a tilt in the anti-inflammatory IL-10 versus pro-inflammatory IL-1β and IL-18 balance during CHIKV nsP-mutant infection that modified acute antiviral and cell signaling canonical pathways. Challenging CHIKV nsP-mutant-infected mice with WT-CHIKV or the closely related O'nyong-nyong virus resulted in no detectable viremia, observable joint inflammation, or damage. Challenged mice showed high antibody titers with efficient neutralizing capacity, indicative of immunological memory. Manipulating molecular processes that govern CHIKV replication could lead to plausible vaccine candidates against alphavirus infection.

**Keywords** chikungunya; live-attenuated vaccine; mutation; neutralizing antibody; non-structural protein
**Subject Categories** Immunology; Microbiology, Virology & Host Pathogen Interaction

## Introduction

Chikungunya virus (CHIKV) is a mosquito-borne alphavirus that causes chikungunya fever (CHIKF). Patients with CHIKF suffer from hallmark bilateral polyarthralgia accompanied by abrupt fever onset, a maculopapular rash, headache, and fatigue (Kam et al, 2009; Schwartz & Albert, 2010). Viremia in the majority of CHIKV-infected patients usually clears after 10 days, with no signs of further complications; however, up to 30% of patients can go on to experience prolonged debilitating arthralgia that can last up to years (Strauss & Strauss, 1994).

CHIKV outbreaks are prevalent, as highlighted by recent outbreaks in Bangladesh in 2017 and the Americas, Thailand, Sudan, and Kenya in 2018 (Kabir et al, 2017; European Centre for Disease Prevention and Control, 2018). Given the acute and chronic disabilities caused by CHIKV infection and the associated economic burden, effective preventive measures are urgently needed to halt CHIKV global expansion. A number of CHIKV vaccine candidates have been described, including attenuated or inactivated CHIKV, alphavirus chimeras, and subunit and genetic vaccines (Eckels et al, 1970; Harrison et al, 1971; Edelman et al, 2000; Wang et al, 2008). Attenuated vaccines can potently trigger CHIKV protection by stimulating both innate and adaptive immunity without causing disease (Baron, 1996). One attenuated CHIKV vaccine (181/clone 25) was developed and clinically evaluated (Levitt et al, 1986; McClain et al, 1998) but clinical trials were halted early when five treated subjects developed transient (< 24 h), mild arthralgia within the first 2 weeks of vaccination (Gorchakov et al, 2012). Recently, a live-attenuated vaccine (MV-CHIK), generated from the insertion of CHIKV structural proteins into measles vectored vaccine strain, showed positive results in a phase 2 clinical trial (Ramsauer et al, 2015; Reisinger et al, 2018).

1 Singapore Immunology Network (SIgN), Agency for Science, Technology and Research (A*STAR), Singapore City, Singapore
2 NUS Graduate School for Integrative Sciences and Engineering, National University of Singapore, Singapore City, Singapore
3 Molecular Microbial Pathogenesis Unit, Department of Cell Biology and Infection, Institute Pasteur, Paris, France
4 Institute of Technology, University of Tartu, Tartu, Estonia
5 School of Biological Sciences, Nanyang Technological University, Singapore City, Singapore
6 Sir William Dunn School of Pathology, University of Oxford, Oxford, UK
7 Fred Hutchinson Cancer Research Center, Seattle, WA, USA
8 Institute of Molecular and Cell Biology, A*STAR, Singapore City, Singapore
9 Department of Biochemistry, Yong Loo Lin School of Medicine, National University of Singapore, Singapore City, Singapore
10 Institute of Infection and Global Health, University of Liverpool, Liverpool, UK
*Corresponding author. Tel: +65 64070028; Fax: +65 64642057; E-mail: lisa_ng@immunol.a-star.edu.sg

 

Innate immunity is the first line of defense against early pathogen infection, as adaptive immunity takes > 1 week to develop. A type-I interferon (IFN) response is induced during CHIKV infection in both patients and experimental models (Ng *et al*, 2009; Gardner *et al*, 2010; Chow *et al*, 2011; Teng *et al*, 2015; Simarmata *et al*, 2016). The importance of this response was demonstrated when mice deficient in type-I IFN receptors succumbed to CHIKV infection (Couderc *et al*, 2008). Another study showed that CHIKV nsP2 inhibited type-I IFN signaling and reduced expression of antiviral mediators in the host by inhibiting IFN-stimulated JAK/STAT signaling and suppressing IFN-induced gene expression (Fros *et al*, 2010). Others have demonstrated the effects of key missense mutations on alphavirus non-structural proteins (nsPs) processing and downstream virus–host interactions (Heise *et al*, 2003; Saul *et al*, 2015). These findings highlight that CHIKV nsPs may be manipulated to generate attenuated CHIKV vaccine candidates.

Here, we characterized the activity of three CHIKV nsP-mutants *in vitro* and *in vivo* and determined the effects of these mutants on CHIKV pathogenesis. We then evaluated whether these CHIKV nsP-mutants may serve as potential CHIKV vaccines. Finally, we monitored whether the vaccine candidate elicited cross-protection with a re-emerging, closely related alphavirus, O'nyong-nyong virus (ONNV).

# Results

## Mutations in non-structural proteins (nsPs) decrease CHIKV infectivity *in vitro*

Missense mutations in the Semliki Forest virus (SFV) nsP1/nsP2 cleavage site and nsP2 protease region influence nsP processing and host immune responses (Saul *et al*, 2015). Using this concept, the corresponding point mutations were generated in CHIKV to determine whether these also elicit an effect on the immune response. The three mutants comprised: (i) RH-CHIKV, an arginine to histidine substitution (R532H) in position P4 of the processing site between nsP1 and nsP2 (nsP1/2 cleavage site); (ii) EV-CHIKV, a glutamic acid to valine substitution (E515V) in nsP2; and (iii) a double RHEV-CHIKV mutant (Fig 1A).

The effects of the mutations on virus infectivity and replication *in vitro* in MTFs were first monitored. Although all three CHIKV nsP-mutants showed decreased virus infectivity capacity (Figs 1B and EV1A), only RH-CHIKV and RHEV-CHIKV showed a low viral RNA titer compared to WT-CHIKV at 12 hpi (Fig EV1B), indicative of reduced viral replication. IFN-α and IFN-β levels in the MTF supernatant were then quantified, as early CHIKV replication in host cells is tightly regulated by the type-I IFN response (Schilte *et al*, 2010). Indeed, the reduced viral RNA titers produced by RH-CHIKV and RHEV-CHIKV were associated with higher soluble IFN-α and IFN-β levels (Fig EV1C and D). Conversely, we observed low levels of IFN-α and IFN-β induction in the WT- and EV-CHIKV-infected MTFs at 12 hpi (Fig EV1C and D), suggesting that these viruses did not induce a strong type-I IFN response.

The attenuation of RH-CHIKV and RHEV-CHIKV could stem from changes in polyprotein processing due to mutation, like in the case of SFV (Saul *et al*, 2015). To analyze the effects of R532H and E515V mutation on CHIKV polyprotein processing, a pulse-chase experiment was performed in BHK-21 cells. For WT-CHIKV-infected cells, all mature nsPs, P1234 precursor, P123, P12, and P34 processing intermediates were detected in pulse samples (0 min; Fig EV1E). The levels of nsPs were shown to increase during the 45-min chase experiment. EV-CHIKV samples also showed comparable nsPs levels during the pulse (0 min) and chase (45 min). In contrast, the amount of mature nsPs in RH-CHIKV- and RHEV-CHIKV-infected cells were significantly decreased in both pulse (0 min) and chase (45 min), indicating a reduction in polyprotein processing efficiency and CHIKV replication.

## RH and RHEV nsP-mutants attenuate CHIKV pathogenesis *in vivo*

To ascertain whether reduced CHIKV replication also affects disease outcomes *in vivo*, CHIKV nsP-mutants were assessed in adult mice (Gardner *et al*, 2010). Compared to the classic WT-CHIKV infection phenotype, a drastic reduction in acute viremia (1–4 dpi) was observed in RH-CHIKV- and RHEV-CHIKV-infected mice (2–3 log units). In addition, complete viremia clearance was achieved by 8 dpi, 2 days earlier than in WT-CHIKV infection (Fig 1C). Nevertheless, the trend of disease progression remained similar between WT-CHIKV, RH-CHIKV, and RHEV-CHIKV, with joint footpad inflammation peaking at 6 dpi (Fig 1D). Instead, EV-CHIKV showed heightened disease progression, with significantly higher viremia and severe joint inflammation compared to WT-CHIKV, RHEV-CHIKV, and RH-CHIKV (Fig 1C and D). These results are in line with the inefficient polyprotein processing of RH-CHIKV and RHEV-CHIKV, affecting replication complex assembly and CHIKV replication.

To examine whether reduced CHIKV nsP-mutant infectivity *in vitro* (Fig 1B) translated *in vivo*, joint footpad cells were harvested from mice during the peak of WT-CHIKV viremia at 2 dpi and infection profiles were determined by flow cytometry (Fig EV2A). All three CHIKV nsP-mutants showed significantly reduced virus infection in joint footpad CD45$^+$ leukocytes at 2 dpi (Fig 1E). In addition, the reduced virus infectivity of all CHIKV nsP-mutants was also observed in CD45-negative cells in the CHIKV-infected joints (Fig EV2B). The reduced infectivity in RH-CHIKV and RHEV-CHIKV could be due to an inefficient replication complex assembly. Interestingly, EV-CHIKV showed comparable pathology in virus-infected joints and a higher viremia as WT-CHIKV, but infectivity in cell subsets was also reduced. It is possible that EV-CHIKV was replicating well in other cell types like fibroblasts (Couderc *et al*, 2008), which contributed to the increased viral titer.

## Suppressed disease pathology during RH-CHIKV and RHEV-CHIKV infection is driven by differential host type-I IFN and acute innate immune responses

To investigate whether differences in the type-I IFN response (IFN-α, IFN-β) could be observed *in vivo*, IFN-α and IFN-β levels were assessed in joint footpad cell lysates at 15 hpi. Consistent with earlier findings *in vitro*, both RH-CHIKV and RHEV-CHIKV infections induced higher IFN-α and IFN-β levels compared to WT-CHIKV and EV-CHIKV (Fig 1F), despite achieving a lower infection level than WT-CHIKV.

The levels of immune mediators in joint footpad cell lysates were next quantified in the early acute infection phase (15 hpi), and at

the peaks of viremia and first joint inflammation (2 dpi) and inflammation (6 dpi; Fig 2). At 15 hpi, pro-inflammatory mediators (IL-1β, TNF-α, IL-15), chemokines (GRO-α, MCP-1, RANTES, MIP-1β, MIP-1α), and hematopoietic growth factors (GM-CSF) were significantly higher in the virus-infected joints of RH-CHIKV and

RHEV-CHIKV mutants relative to WT-CHIKV (Fig 2, Dataset EV1). These findings are consistent with the high levels of IFN-α and IFN-β in these mutants (Fig 1F). Higher levels of anti-inflammatory cytokines IL-4 and IL-5 were also observed in these mutants compared to WT-CHIKV (Fig 2, Dataset EV1).

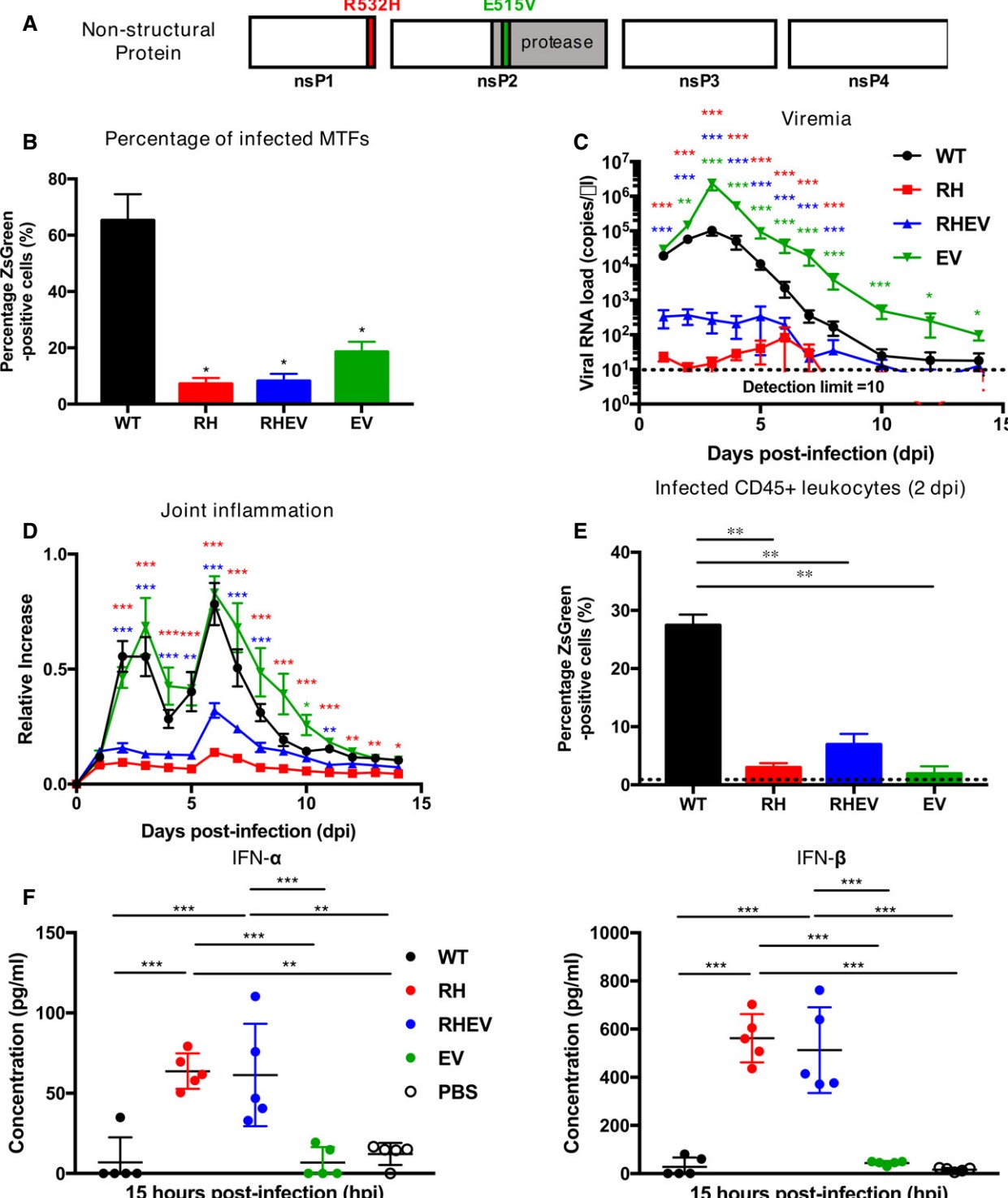

Figure 1.

◄

**Figure 1.   R532H and E515V mutations in the nsP region limit CHIKV pathogenicity and alter type-I interferon responses in WT C57BL/6 mice.**

A    Schematic of the CHIKV mutation sites, including CHIKV with R to H (RH), E to V (EV), or both (RHEV) amino acid (a.a) substitutions.
B    Percentage infectivity of mouse tail fibroblasts (MTFs) infected with WT-, RH-, RHEV-, and EV-CHIKV (MOI 10, *n* = 3 per group) harvested at 12 hpi. Data are presented as the means ± SEM. Statistical analyses were performed using two-tailed paired *t*-test (*P = 0.019 RH-CHIKV, *P = 0.019 RHEV-CHIKV, *P = 0.020 EV).
C, D  (C) Viremia progression and (D) joint inflammation in virus-infected mice were monitored over 2 weeks (*n* = 11 per group). Viremia is detected with CHIKV nsP1 probe via qRT–PCR. Statistical analyses were performed using two-tailed Mann–Whitney *U*-test; viremia (***P = 0.000003 RH-CHIKV 1–5 dpi, ***P = 0.000065 RH-CHIKV 6 dpi, ***P = 0.000281 RH-CHIKV 7 dpi, ***P = 0.000079 RH-CHIKV 8 dpi, ***P = 0.000275 RHEV-CHIKV 1 dpi, ***P = 0.000281 RHEV-CHIKV 2–4 dpi, ***P = 0.000054 RHEV-CHIKV 5 dpi, ***P = 0.000763 RHEV-CHIKV 6 dpi, ***P = 0.000077 RHEV-CHIKV 7 dpi, ***P = 0.000595 RHEV-CHIKV 8 dpi, **P = 0.007 EV-CHIKV 2 dpi, ***P = 0.000011 EV-CHIKV 3 dpi, ***P = 0.000034 EV-CHIKV 4 dpi, ***P = 0.000394 EV-CHIKV 5 dpi, ***P = 0.000128 EV-CHIKV 6 dpi, ***P = 0.000006 EV-CHIKV 7 dpi, ***P = 0.000054 EV-CHIKV 8 dpi, ***P = 0.00017 EV-CHIKV 10 dpi, *P = 0.041 EV-CHIKV 12 dpi, *P = 0.012 EV-CHIKV 14 dpi) and joint inflammation (***P = 0.000003 RH-CHIKV 2–4 dpi, ***P = 0.000006 RH-CHIKV 5 dpi, ***P = 0.000003 RH-CHIKV 6–8 dpi, ***P = 0.000085 RH-CHIKV 9 dpi, ***P = 0.000045 RH-CHIKV 10 dpi, ***P = 0.000011 RH-CHIKV 11 dpi, **P = 0.002 RH-CHIKV 12 dpi, **P = 0.004 RH-CHIKV 13 dpi, *P = 0.012 RH-CHIKV 14 dpi, ***P = 0.000003 RHEV-CHIKV 2–3 dpi, ***P = 0.000411 RHEV-CHIKV 4 dpi, **P = 0.008 RHEV-CHIKV 5 dpi, ***P = 0.000017 RHEV-CHIKV 6 dpi, ***P = 0.000094 RHEV-CHIKV 7 dpi, ***P = 0.000964 RHEV-CHIKV 8 dpi, **P = 0.002 RHEV-CHIKV 11 dpi, ***P = 0.029 EV-CHIKV 10 dpi). Error bars are SEM.
E    Compiled percentages of CHIKV-infected (ZsGreen positive) leukocytes at 2 days post-infection (dpi) for the respective groups (*n* = 5 per group). Background ZsGreen signals are indicated by the dotted line. Statistical analyses were performed using two-tailed Mann–Whitney *U*-test **P = 0.008 RH-CHIKV, **P = 0.008 RHEV-CHIKV, **P = 0.008 EV-CHIKV. Error bars are SD.
F    Dot-plots of the localized type-I interferon (IFN-α and IFN-β) concentrations at 15 hpi. Data are expressed as the means ± SD. Data were analyzed by one-way ANOVA with *post hoc* Tukey test; IFN-α (***P = 0.000418 WT-CHIKV versus RH-CHIKV, ***P = 0.000688 WT-CHIKV versus RHEV-CHIKV, ***P = 0.000407 RH-CHIKV versus EV-CHIKV, **P = 0.00122 RH-CHIKV versus PBS, ***P = 0.00067 RHEV-CHIKV versus EV-CHIKV, **P = 0.002014 RHEV-CHIKV versus PBS) and IFN-β (***P < 0.000001 WT-CHIKV versus RH-CHIKV, ***P < 0.000001 WT-CHIKV versus RHEV-CHIKV, ***P < 0.000001 RH-CHIKV versus EV-CHIKV, ***P < 0.000001 RH-CHIKV versus PBS, ***P = 0.000001 RHEV-CHIKV versus EV-CHIKV, ***P < 0.000001 RHEV-CHIKV versus PBS). See also Fig EV1. *n* = 5 per group. Data is from one independent experiment.

At 2 dpi, this induction of immune mediators in the RH-CHIKV- and RHEV-CHIKV-infected joints became less prominent. Instead, cytokines such as GM-CSF, G-CSF, IL-4, IL-5, and IL-18 were increased in the virus-infected joints of WT-CHIKV and EV-CHIKV mice (Fig 2, Dataset EV1), suggesting a slower inflammatory response compared to RH-CHIKV- and RHEV-CHIKV-infected joints. At the peak of joint inflammation (6 dpi), RH-CHIKV- and RHEV-CHIKV-infected joints showed higher levels of anti-inflammatory mediator IL-10, and the chemokine eotaxin. Importantly, these infected joints also showed lower levels of pro-inflammatory mediators (IL-1β, IL-1α, IL-6, IL-23, TNF-α) and higher levels of pro-inflammatory mediators (IL-17A, IL-31) when compared to WT-CHIKV-infected joints (Fig 2, Dataset EV1). The well-regulated immune responses in the RH-CHIKV- and RHEV-CHIKV-infected mice indicate that an early increase in pro-inflammatory mediators (15 hpi) followed by gradual depletion from 2 to 6 dpi is associated with suppressed disease pathology.

**Differences in acute antiviral and cell signaling canonical pathways augment CHIKV disease pathology**

To identify the possible cellular pathways involved in the suppressed CHIKV disease pathology elicited by attenuated CHIKVs (RH-CHIKV and RHEV-CHIKV), ingenuity pathway analysis (IPA) was performed on the analytes described above (Fig 2) and on publicly available RNA-seq data from WT-CHIKV-infected joints (S1 Table in Wilson *et al*, 2017) exhibiting complete or partial agreement (either 0 dpi to 2 dpi, or 2 dpi to 6/7 dpi) in their expression state.

IPA of the analytes and transcripts that were up-regulated from mock to 2 dpi and then down-regulated from 2 dpi to 6–7 dpi (as previously observed; Fig 2) was first performed. Four of the top five canonical pathways found were involved in acute antiviral responses (Fig 3B). Many of the associated differentially expressed transcripts present in the acute antiviral canonical pathways, including DDX58, DHX58, IFIH1, IFIT2, IFNA4, IFNB1, IL5, IL6, IL10, IRF7, ISG15, STAT2, CCL5, EIF2AK2, LIF, OAS1, OAS2, OAS3,

TLR2, TLR3, IFIT3, and IFITM3 (Fig 3C), have been previously associated with the acute phase of CHIKV infection (Sourisseau *et al*, 2007; Fros *et al*, 2010; Gardner *et al*, 2010; Werneke *et al*, 2011; Schilte *et al*, 2012; Chiam *et al*, 2015; Choi *et al*, 2015; Her *et al*, 2015; Reynaud *et al*, 2015; Inglis *et al*, 2016; Poddar *et al*, 2016; Sanchez David *et al*, 2016; Simarmata *et al*, 2016). Comparing all CHIKV nsP-mutants with WT-CHIKV-infected joints revealed differences in GRO-α at 2 dpi, and IL-27 and IL-10 at 6 dpi in RH-CHIKV-infected joints, which participate in these acute antiviral canonical pathways (Fig 3A).

IPA was then repeated for analytes and transcripts that were up-regulated from mock to 2 dpi, but showed no difference from 2 dpi to 6–7 dpi (Fig 3D). Here, two of the top five canonical pathways were involved in cell signaling (Fig 3E). Many of the associated transcripts in the cell signaling canonical pathways, including CCL4, CD40, CSF2, IL15, IL18, IL1β, and TNF (Fig 3F), have been previously associated with CHIKV infection (Ng *et al*, 2009; Labadie *et al*, 2010; Chow *et al*, 2011; Kelvin *et al*, 2011; Simarmata *et al*, 2016; Michlmayr *et al*, 2018). Comparing all CHIKV nsP-mutants with WT-CHIKV-infected joints revealed differences in analytes involved in cell signaling pathways, including MIP-1β, IL-1β, IL-18, IFN-γ, and TNF-α in RH-CHIKV-infected joints (Fig 3D).

**High-dimensional analysis reveals phenotypic differences in joint-infiltrating leukocytes between CHIKV nsP-mutant mice**

The differences observed in the immune mediator profiles in all CHIKV nsP-mutant-infected groups may be due to differences in the leukocyte profile infiltrating the joints. To investigate this, joint foot-pad cells were harvested at 6 dpi and the immune-cell profiles were characterized by flow cytometry (Fig EV3A). After quantifying the cell counts from major leukocyte subsets, including neutrophils, LFA-1[+] CD4[+] T cells, LFA-1[+] CD8[+] T cells, natural killer (NK) cells, and macrophages, no significant differences between the different groups were observed (Figs 4A and EV3A).

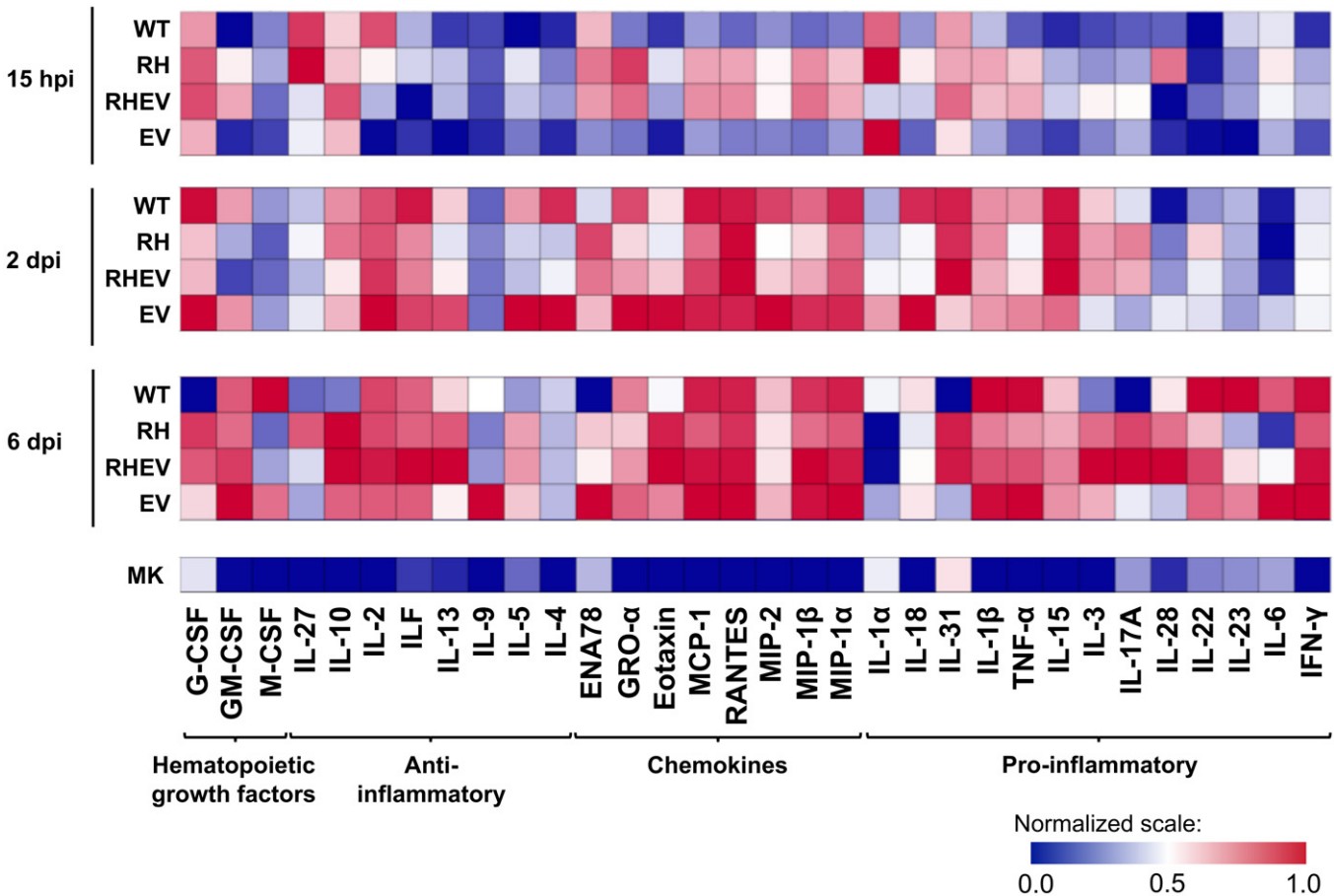

**Figure 2. Mutations in CHIKV nsP regions affect local immune responses during the acute disease phase.**

Cell lysates were collected from the joints of mice infected with WT-CHIKV or CHIKV nsP-mutants at 15 hpi, 2 dpi, or 6 dpi (*n* = 5 per group). The concentrations of 36 immune mediators were quantified using a 36-plex microbead-based immunoassay. Immune mediators are grouped based on function. Each color represents the relative concentration of a particular analyte. Blue and red indicate low and high concentrations, respectively. See also Dataset EV1.

To analyze this high-dimensional cytometry dataset in an unbiased way, UMAP [an algorithm similar to t-SNE that reduces data into two dimensions to retain both local and global structures (Becht *et al*, 2018; preprint: McInnes & Healy, 2018)] and Pheno-Graph were used to obtain 33 clusters of cells with homogeneous marker expression profiles (Levine *et al*, 2015). Cluster identity should be based on the collective expression levels of lineage surface markers. Using UMAP, a dissimilarity metric was computed across the samples (Jensen–Shannon divergence; Amir *et al*, 2013), and it confirmed that immune cells of each virus-infected group were more similar to one another than to mock-infected samples (Fig 4B). This result is consistently observed on the UMAP biplot, which showed that mock-infected joints were lacking an infiltrating leukocyte population corresponding to PhenoGraph clusters 1, 13, 14, 19, 29, 30, 32, 33 (Fig 4C–E). These clusters are collectively made up of populations expressing $CD11b^+$, $CD64^+$, $Ly6C^{hi}$, and $MHC-II^+$ (Figs 4E and EV3B), indicative of inflammatory monocytes (Haist *et al*, 2017). Interestingly, the RH-CHIKV-infected group diverged the most from WT-CHIKV and showed the highest level of similarity to the mock-infected group (Fig 4B).

**Challenge with RH- and RHEV-CHIKVs induces long-lasting, neutralizing CHIKV-specific antibodies**

Given the attenuated infection phenotype elicited by RH-CHIKV and RHEV-CHIKV *in vivo*, we hypothesized that these viruses may have potential as CHIKV vaccines. We considered the first infection as an episode of vaccination (Fig 1C and D) and assessed the quantity and quality of the antibodies generated in these mice using virion-based ELISA and neutralization assays (Kam *et al*, 2012a). While all mice vaccinated with CHIKV produced specific IgG detectable from 6 days post-vaccination (dpv), the levels produced following RH-CHIKV and RHEV-CHIKV infection were lower than WT-CHIKV and EV-CHIKV (Fig 5A). As the infection progressed to 14 dpv, however, we detected similar levels of CHIKV-specific IgG between all infected groups (Fig 5A). Concordantly, antibodies generated by RH-CHIKV- and RHEV-CHIKV-vaccinated mice were able to neutralize CHIKV infection from 6 dpv (Fig EV5A).

To qualify as a protective CHIKV vaccine, the vaccine candidates should induce lasting CHIKV neutralizing antibodies. Indeed, all three CHIKV nsP-mutant-vaccinated groups showed similar titers of CHIKV-specific IgG at 90 dpv (Fig 5A). Half maximal inhibitory

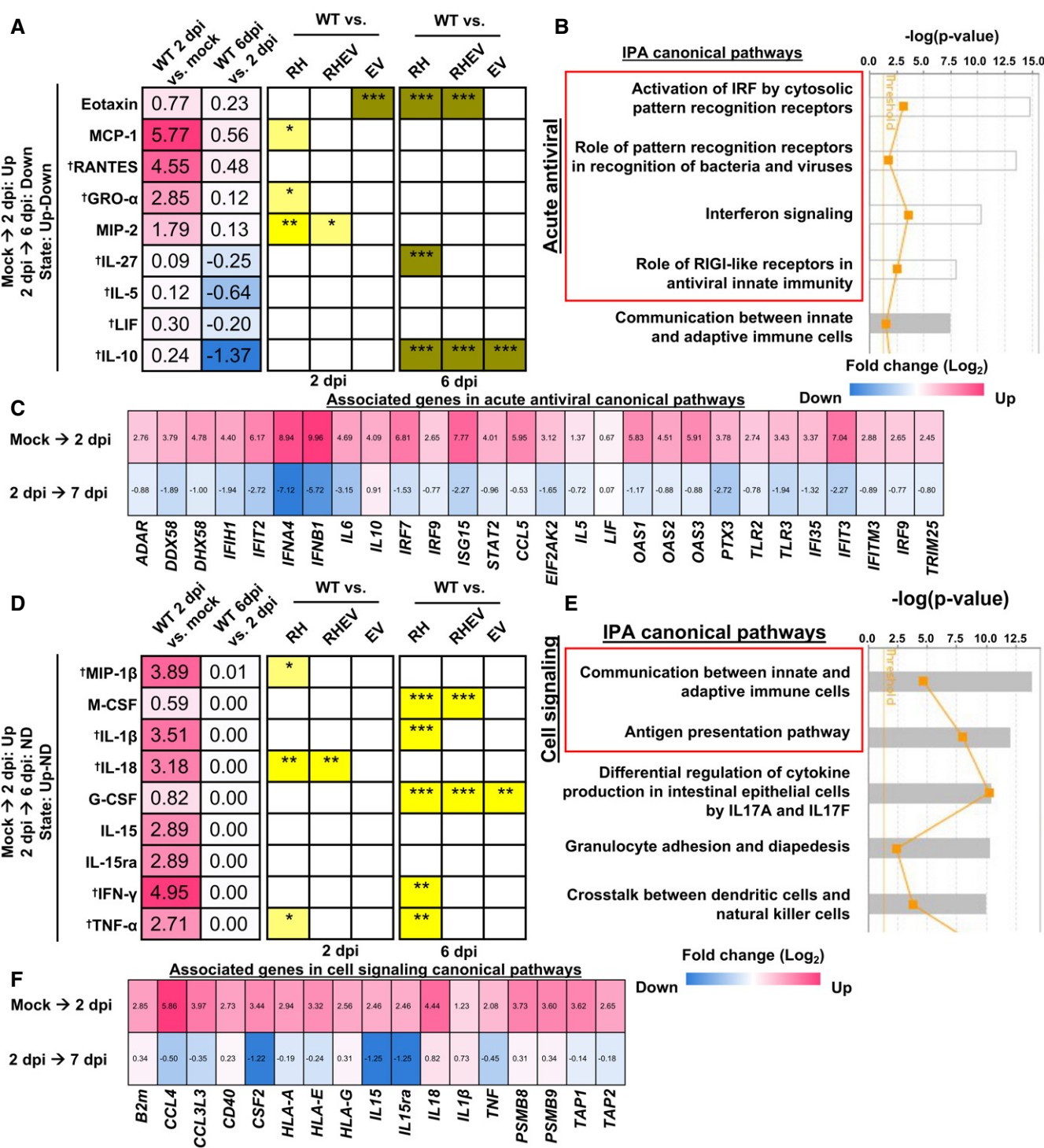

**Figure 3. Attenuated CHIKV infection may be due to aberrations in acute antiviral and cell signaling canonical pathways.**

A–F Luminex analytes from WT-CHIKV-infected joint cell lysates are categorized based on fold change states [up, down, or no difference (ND)] at 2 dpi relative to mock, and 6 dpi relative to 2 dpi. The numbers in the heatmap represent Log$_2$ fold changes of analytes or genes. Analytes in complete or partial agreement with corresponding gene expression values derived from RNA-sequencing (RNA-seq) data appeared predominantly in the (A) up then down state (mock to 2 dpi, then 2 to 6 dpi, respectively), or (D) up then ND state (mock to 2 dpi, then 2 to 6 dpi, respectively). † indicates analyte involvement in the canonical pathways. Differences in analyte concentrations during attenuated CHIKV infection are indicated in the checkbox. Yellow represents lower, while green represents higher concentrations relative to WT-CHIKV. Data were analyzed by one-way ANOVA with *post hoc* Tukey test; *$P < 0.05$, **$P < 0.01$, and ***$P < 0.001$. (B, E) Top canonical pathways discovered by ingenuity pathway analysis (IPA). The study analytes and transcripts in the RNA-seq data from wild-type CHIKV-infected and mock-infected samples in the same states were used. (C, F) Corresponding associated transcripts in acute antiviral and cell signaling canonical pathways are presented as a heatmap showing the Log$_2$ fold change. For exact *P*-values, see Dataset EV2.

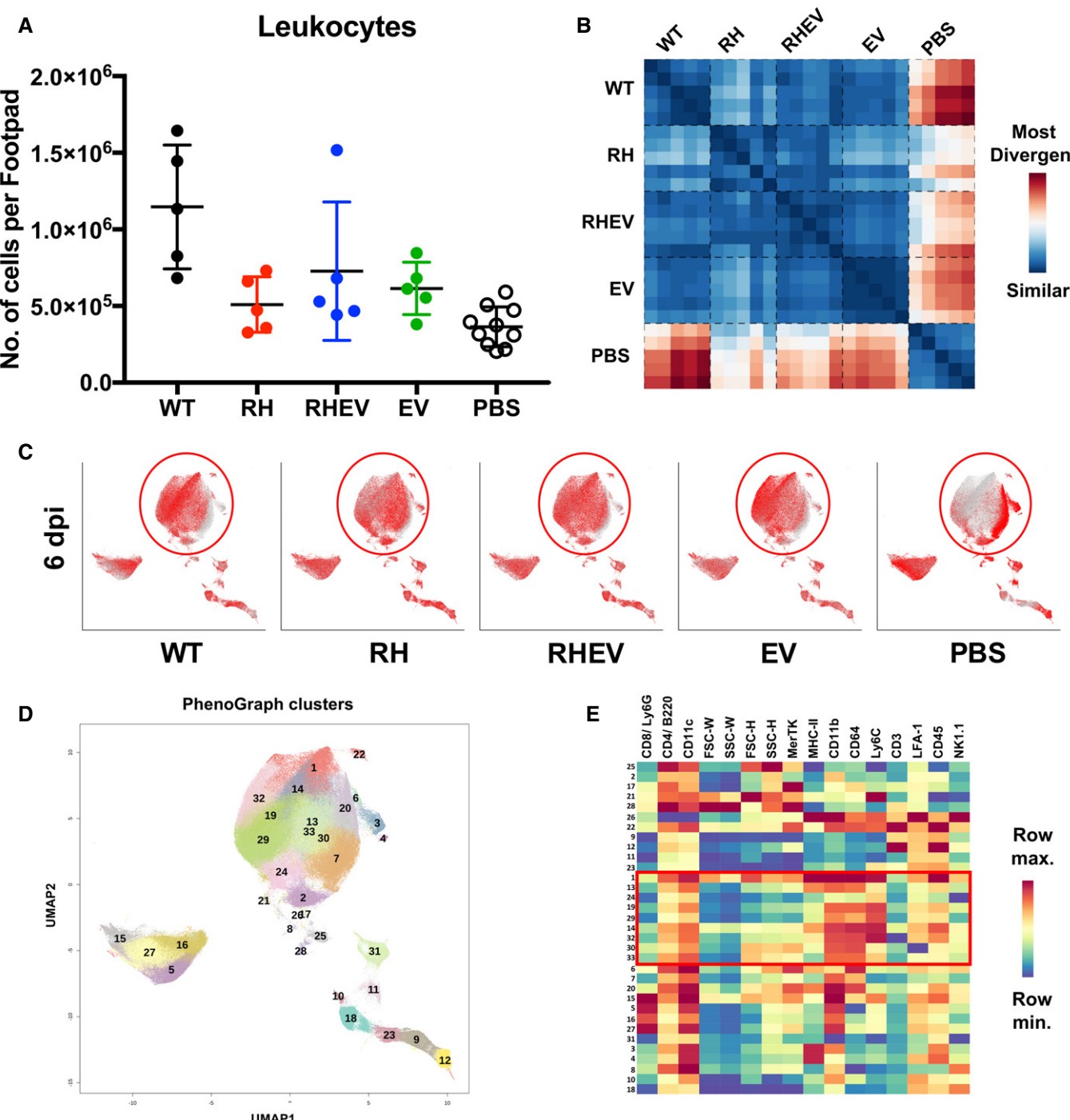

**Figure 4.  High-dimensional analysis of fluorescence-activated cell sorting data reveals differences in CHIKV nsP-mutant-infected groups.**

Joints from WT-CHIKV-, RH-CHIKV-, RHEV-CHIKV-, EV-CHIKV-, and mock-infected WT C57BL/6 mice were harvested at 6 dpi.

A   Absolute numbers of leukocytes present in the joints as determined by immune-phenotyping. Data are presented as the means $\pm$ SD ($n$ = 5 per group, except PBS $n$ = 10). No significance was obtained between the groups by Kruskal–Wallis test with Dunn's multiple comparisons.

B   Jensen–Shannon divergence heatmap showing the level of divergence between infected groups.

C   Superimposed PhenoGraphs of UMAP transformed FACS data from immune-phenotyping of joint cells from WT-CHIKV-, RH-CHIKV-, RHEV-CHIKV-, EV-CHIKV-, and mock-infected animals ($n$ = 5 per group).

D   PhenoGraph cluster IDs from 1 to 33. Clusters are grouped in proximity based on surface marker phenotypic similarities and do not represent conventional cell types.

E   Cluster ID annotation with heatmap. Cluster IDs are indicated in rows, while surface marker expression levels are indicated by the columns. The color represents the relative mean fluorescence intensity (MFI) of a surface marker. Blue and red represent low and high MFI, respectively. Red square and ovals highlight differences of several PhenoGraph clusters in infected joints at 6 dpi. See also Fig EV3.

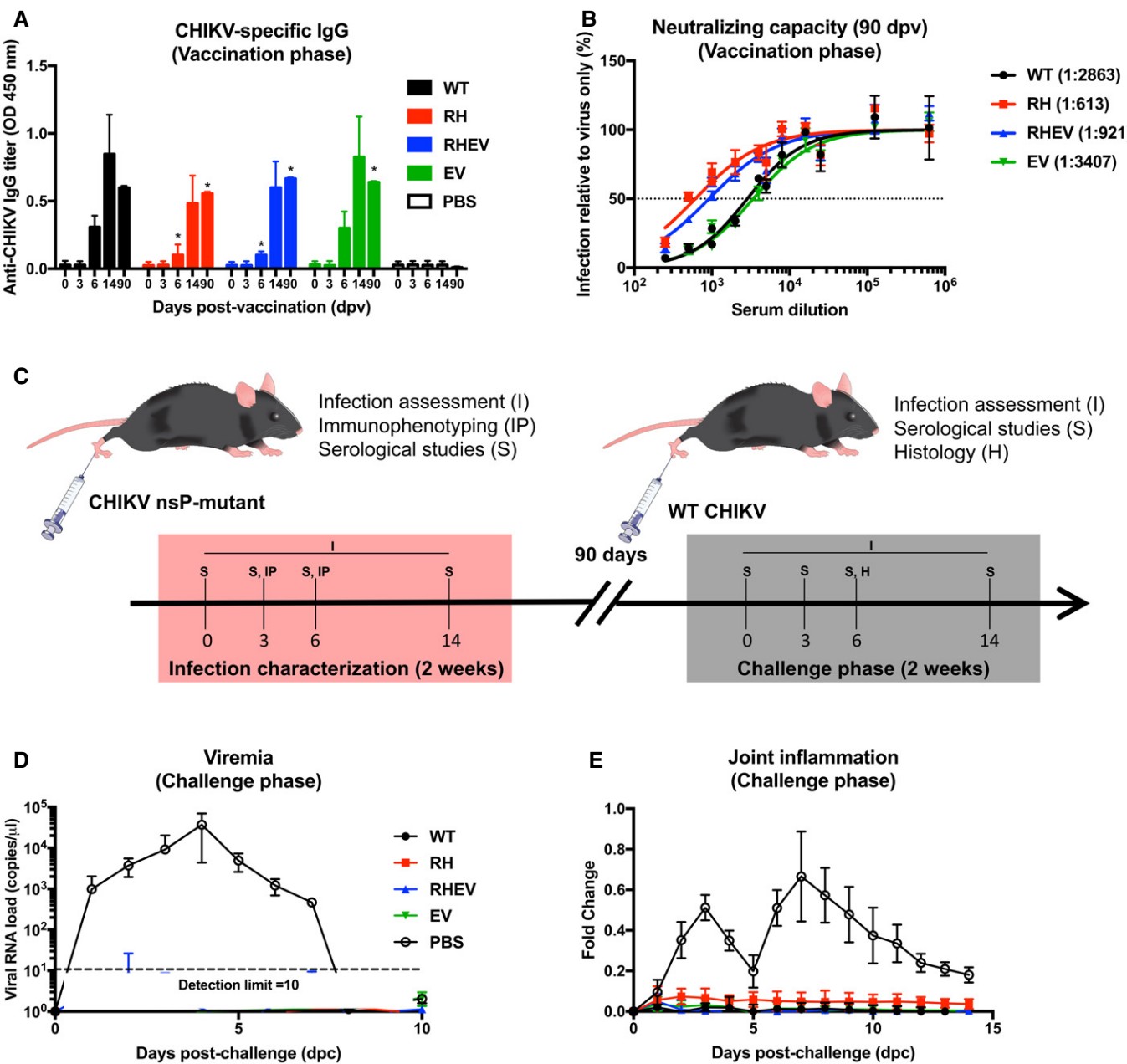

**Figure 5. Vaccination with CHIKV nsP-mutants induces neutralizing antibodies and protects against pathology during CHIKV challenge.**

A   Anti-CHIKV IgG titers from the vaccination phase were determined with a CHIKV virion-based ELISA. Sera from four animals were pooled and diluted 1:500 for the assay. Statistical analyses were performed using two-tailed paired $t$-test (*$P$ = 0.029 RH-CHIKV 6 dpv, *$P$ = 0.029 RH-CHIKV 90 dpv, *$P$ = 0.029 RHEV-CHIKV 6 dpv, *$P$ = 0.029 RHEV-CHIKV 90 dpv, *$P$ = 0.029 EV-CHIKV 90 dpv). Error bars are in SD.

B   Half maximal inhibitory concentration ($IC_{50}$) of pooled sera collected on 90 dpv was characterized in HEK293T cells infected with WT-CHIKV (MOI 5). Sera were serially diluted at 1:625,000, 1:125,000, 1:25,000, 1:16,000, 1:8,000, 1:5,000, 1:4,000, 1:2,000, 1:1,000, 1:500, 1:250. Sera dilution to obtain $IC_{50}$ is shown in parentheses. Data shown are representative of three independent experiments. Error bars are in SD.

C   WT C57BL/6 mice were vaccinated subcutaneously with 1E6 PFU of WT-CHIKV or CHIKV nsP-mutants (RH, RHEV, and EV) at the metatarsal region of the footpad. Challenge with 1E6 PFU of WT-CHIKV was performed via the same route at 90 dpv.

D, E   (D) Viremia and (E) severity of joint inflammation were monitored over 2 weeks. Viremia is detected with CHIKV nsP1 probe via qRT–PCR. The data are presented as the means ± SD and are representative of two independent experiments ($n$ = 6 per group).

concentration ($IC_{50}$) of the sera collected prior to WT-CHIKV challenge at 90 dpv was also investigated. Notably, these antibodies retained their neutralizing capacity (Fig 5B). Although the neutralizing capacity of antibodies for RH-CHIKV- and RHEV-CHIKV- was

lower than WT-CHIKV-vaccinated mice at 90 dpv, shown by the lower dilution levels to reach $IC_{50}$ [RH-CHIKV (1:613) < RHEV-CHIKV (1:921) < WT-CHIKV (1:2,863)], the levels of CHIKV neutralizing antibodies induced are sufficient for protection *in vivo* when

the vaccinated mice were challenged with WT-CHIKV (Fig 5C). Notably, there was neither detectable viremia in the peripheral blood nor observable joint swelling within 2 weeks from WT-CHIKV challenge in any of the vaccine groups (Fig 5D and E).

As RH-CHIKV has the least pathologic symptoms upon vaccination, and can confer a protective response upon WT-CHIKV challenge, it was investigated further as a vaccine candidate in μMT mice. These mice lack the ability to produce mature B cells and thus are unable to produce CHIKV-specific antibodies following vaccination. As expected, vaccination with WT-CHIKV or RH-CHIKV conferred no protection (Fig EV4A and B), as both groups displayed viremia and joint inflammation during WT-CHIKV challenge (Fig EV4C and D), confirming the role of neutralizing antibodies in disease protection (Lum et al, 2013).

To assess whether WT-CHIKV challenge affected immunological memory development in the different mutant CHIKV-vaccinated mice, we assessed CHIKV-specific IgGs for neutralizing capacity on different days post-challenge (dpc). Serum CHIKV-specific IgGs from all vaccinated mice were significantly higher than mock-vaccinated mice on 3, 6, and 14 dpc (Fig EV5B). In addition, the sera from the vaccinated mice were more neutralizing than the sera obtained from mock-vaccinated mice from 3 dpc (Fig EV5C). In summary, these data show that CHIKV nsP-mutant induced similar CHIKV-specific antibody levels with comparable neutralizing capacity to WT-CHIKV-vaccinated mice (Fig EV5B and C).

### Vaccination with attenuated CHIKV protects against CHIKV-induced tissue damage and edema in the joints

The effects of RH-CHIKV-mediated protection were assessed during WT-CHIKV challenge microscopically in transverse sections of the virus-infected joints (Fig 6). Histologically, the challenged mock-vaccinated mice showed distension of the joint footpad at 6 dpc, characterized by a moderate degree of edema and inflammation. In addition, they exhibited myositis, synovitis, and subcutaneous inflammation around the joint as well as mild-to-moderate muscle degeneration/necrosis (Fig 6A and B). Affected muscle fibers were swollen and irregular in shape, and showed homogeneous loss of cross striations.

In contrast, the joints of WT-CHIKV-vaccinated mice exhibited minimal histopathological features upon challenge (Fig 6A). There was a low degree of edema in subcutaneous tissues and minimal mononuclear cell infiltration in the proximity of muscle bundles. Surprisingly, a moderate number of regenerated muscle fibers were common in this group; a similar degree of microscopic lesions was observed in RH-CHIKV-vaccinated mice, but only a few skeletal muscle fibers showed regenerative changes (Fig 6A and B). The

difference in regenerated muscle fibers could be due to higher extent of muscle damage in mice vaccinated with WT-CHIKV. Mild synovial hyperplasia was more common in the RH-CHIKV-vaccinated group during challenge.

### Vaccinated mice show cross-protection against O'nyong-nyong virus (ONNV) infection

O'nyong-nyong virus is a closely related alphavirus to CHIKV that causes similar, but milder clinical symptoms in patients. Concordantly, mild joint inflammation and low levels of viremia in our in vivo mouse models of ONNV infection were also observed (Fig 7A and B). To investigate whether vaccination with RH-CHIKV offered cross-protection against ONNV infection, vaccinated mice were challenged with WT-ONNV 3 months later and monitored for signs of pathology. Vaccinated mice did not show any detectable viremia or observable footpad inflammation throughout the duration of the ONNV challenge (Fig 7C and D). Of note, ONNV infection in naïve mice 3 months later caused milder joint pathologies compared to 3-week-old mice, and viremia is naturally controlled to a hovering level near the detection limit (Fig 7C and D).

## Discussion

CHIKV has the potential for global transmission; the substantial health complications of CHIKV and resulting economic burden make finding prophylactic and therapeutic strategies against CHIKV infection a high priority. Existing live-attenuated vaccines for poliovirus, yellow fever virus, and measles successfully prevent infection and disease in susceptible populations (Minor, 2015). Live-attenuated vaccines might, therefore, be an effective prophylactic strategy against CHIKV. Recently, MV-CHIK, a live-attenuated recombinant vaccine, showed promising results in phase 2 clinical trials (Ramsauer et al, 2015; Reisinger et al, 2018). However, the functionality of MV-CHIK has only been demonstrated in in vitro assays. Moreover, it is a recombinant vaccine harboring only CHIKV structural genes. As previous studies have shown that antibodies against CHIKV nsP3 are present in CHIKV patient cohorts (Kam et al, 2012a), it is still uncertain how efficacious this vaccine will be in the long term. Here, we found that RH-CHIKV produces a minimal pathologic response upon infection but offers a high level of protection against CHIKV challenge, suggesting that RH-CHIKV presents a plausible preventive measure that warrants further exploration.

This report demonstrates that R532H mutation in nsP1 causes inefficient polyprotein processing and suppresses CHIKV-mediated

▶

**Figure 6. Vaccination with attenuated CHIKVs protects against CHIKV-induced joint pathology.**

A  Representative hematoxylin and eosin (H&E) images of inflamed joint footpad at 6 dpc. Ed, region of edema; red arrow, synovitis; black arrow, normal synovial membrane; blue arrow, mild synovial hyperplasia; D, degeneration of muscle; N, necrosis of muscle; R, regeneration of muscle.

B  Histopathological scoring of edema, inflammation in different regions of the joint footpad, and muscle pathology of WT-CHIKV challenged animals (n = 5) at 6 dpc, performed blind. Three sections from each joint footpad were scored. The data are expressed as the means ± SD. Data were analyzed by one-way ANOVA with Tukey post-test; edema (***P = 0.000002 WT-CHIKV versus PBS, ***P = 0.000004 RH-CHIKV versus PBS), muscle necrosis (**P = 0.005521 WT-CHIKV versus PBS, *P = 0.02628 RH-CHIKV versus PBS), muscle regeneration (***P < 0.000001 WT-CHIKV versus PBS, ***P < 0.000001 WT-CHIKV versus RH-CHIKV), muscle inflammation (*P = 0.040258 WT-CHIKV versus PBS, *P = 0.040258 RH-CHIKV versus PBS), synovial membrane inflammation (*P = 0.014457 WT-CHIKV versus PBS, *P = 0.014457 RH-CHIKV versus PBS), and subcutaneous area inflammation (***P = 0.00004 WT-CHIKV versus PBS, ***P = 0.000074 RH-CHIKV versus PBS).

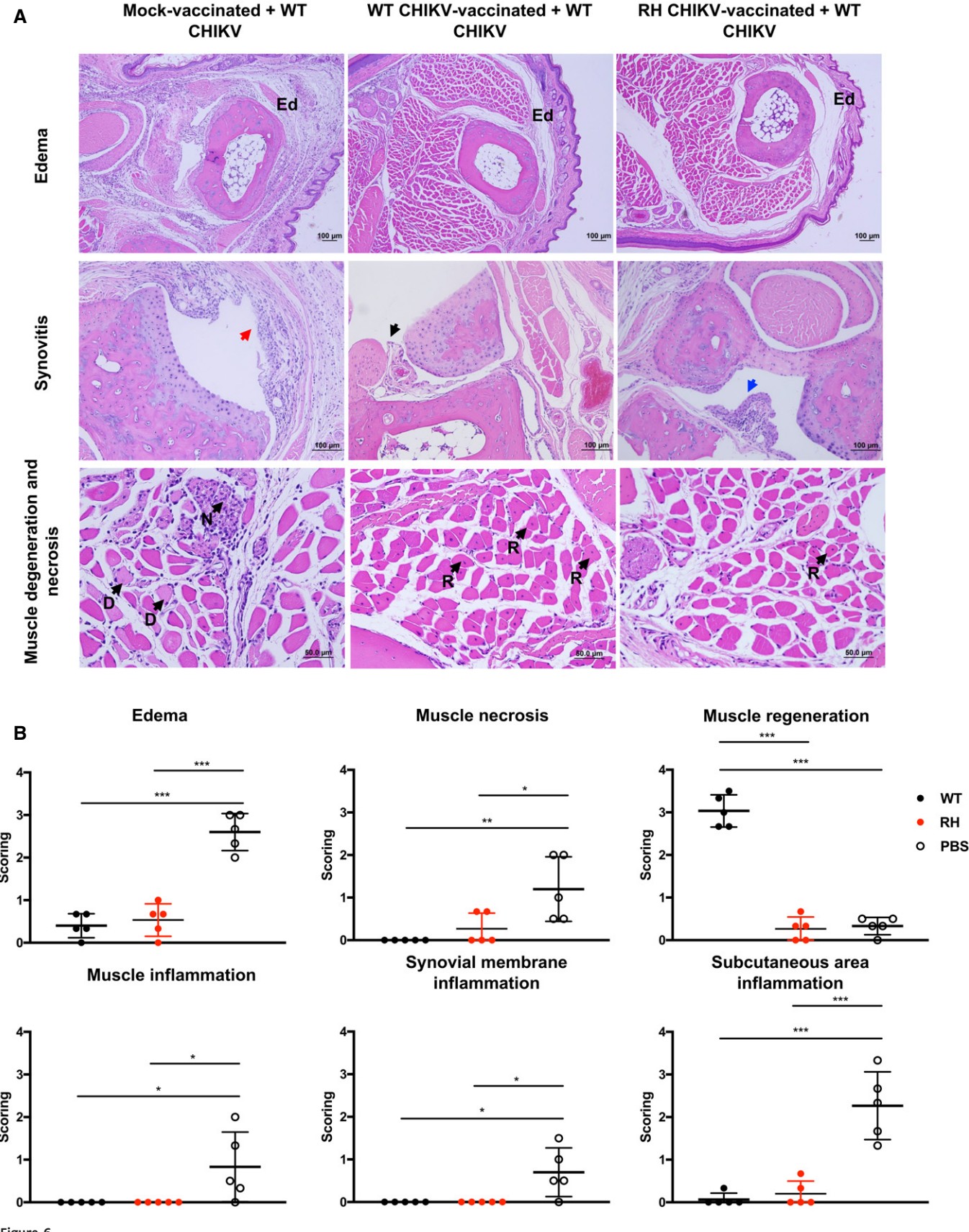

Figure 6.

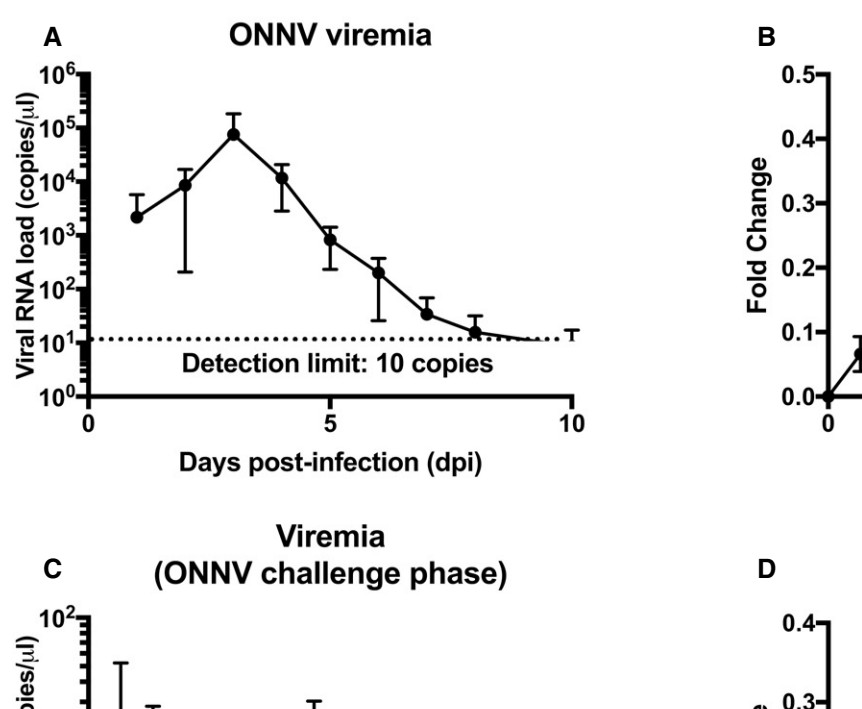

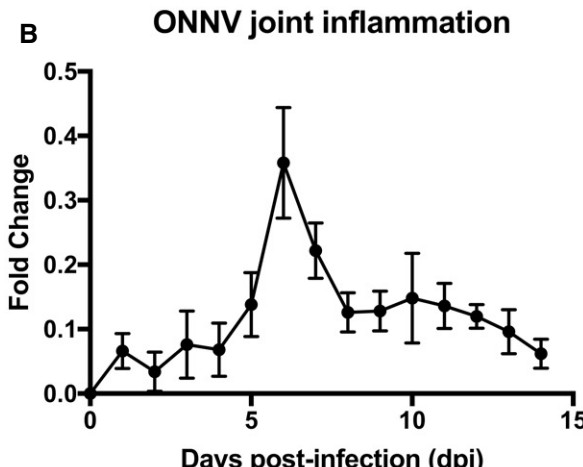

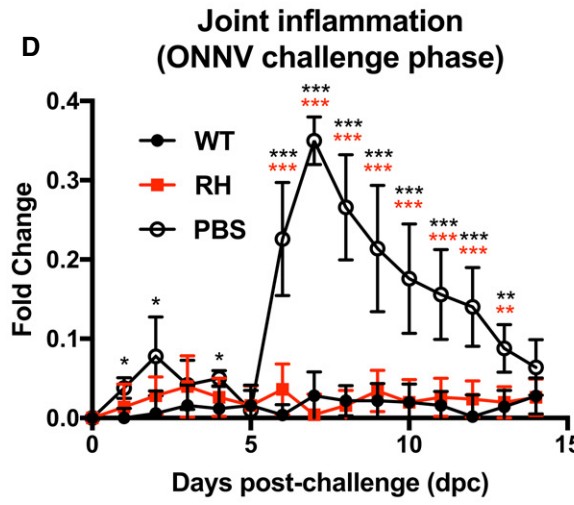

**Figure 7. Vaccination with attenuated CHIKV protects mice against WT O'nyong-nyong virus (ONNV).**

Three-week-old WT C57BL/6 mice were infected with 1E6 PFU of WT-ONNV at the metatarsal region of the footpad.

A, B    (A) Viremia progression and (B) joint inflammation in ONNV-infected mice were monitored over 2 weeks (*n* = 5). Viremia is detected with ONNV nsP1 probe via qRT–PCR. Error bars are in SD.

C, D    (C) Three-week-old WT C57BL/6 mice were vaccinated subcutaneously with 1E6 PFU of WT-CHIKV or attenuated CHIKV (RH) at the metatarsal region of the footpad. Challenge with 1E6 PFU of WT-ONNV was performed via the same route at 3 months post-vaccination. Viremia progression and (D) joint inflammation of the mice were monitored over 2 weeks (*n* = 5 per group). Viremia is detected with ONNV nsP1 probe via qRT–PCR. All data are presented as the means ± SD. Data were analyzed by one-way ANOVA with Tukey post-test; joint inflammation (ONNV challenge phase) (\**P* = 0.027516 WT-CHIKV versus PBS 1 dpc, \**P* = 0.020452 WT-CHIKV versus PBS 2 dpc, \**P* = 0.048344 WT-CHIKV versus PBS 4 dpc, \*\*\**P* = 0.000017 WT-CHIKV versus PBS 6 dpc, \*\*\**P* = 0.000077 RH-CHIKV versus PBS 6 dpc, \*\*\**P* < 0.000001 WT-CHIKV versus PBS 7 dpc, \*\*\**P* < 0.000001 RH-CHIKV versus PBS 7 dpc, \*\*\**P* = 0.000002 WT-CHIKV versus PBS 8 dpc, \*\*\**P* = 0.000002 RH-CHIKV versus PBS 8 dpc, \*\*\**P* = 0.000155 WT-CHIKV versus PBS 9 dpc, \*\*\**P* = 0.000278 RH-CHIKV versus PBS 9 dpc, \*\*\**P* = 0.000411 WT-CHIKV versus PBS 10 dpc, \*\*\**P* = 0.000411 RH-CHIKV versus PBS 10 dpc, \*\*\**P* = 0.000181 WT-CHIKV versus PBS 11 dpc, \*\*\**P* = 0.000351 RH-CHIKV versus PBS 11 dpc, \*\*\**P* = 0.000139 WT-CHIKV versus PBS 12 dpc, \*\*\**P* = 0.000684 RH-CHIKV versus PBS 12 dpc, \*\**P* = 0.001081 WT-CHIKV versus PBS 13 dpc, \*\**P* = 0.002107 RH-CHIKV versus PBS 13 dpc). Black asterisks indicate significant differences between WT and PBS vaccination, while red asterisks indicate differences between RH and PBS vaccination.

joint pathology due to differences in the local immune response. IPA of immune mediators in joint footpad cell lysates combined with differential gene expression data for CHIKV-infected mice taken from publicly available RNA-seq data (S1 Table in Wilson *et al*, 2017) identified several possible canonical pathways that may influence CHIKV severity during WT-CHIKV infection. Notably, several of the transcripts were linked to the acute antiviral canonical pathways that are functionally important for CHIKV sensing (DDX58, DHX58,

TLR2, TLR3, OAS3; Bréhin *et al*, 2009; Her *et al*, 2015; Inglis *et al*, 2016; Sanchez David *et al*, 2016), type-I IFN responses (IFNA4, IFNB1, STAT2, IRF7, ISG15, IFIH1, IFITM3; Fros *et al*, 2010; Gardner *et al*, 2010; Poddar *et al*, 2016; Sanchez David *et al*, 2016; Schilte *et al*, 2012; Werneke *et al*, 2011), and inhibiting CHIKV replication (EIF2AK2, IFIT2, IFIT3; Chiam *et al*, 2015; Reynaud *et al*, 2015). These findings support that acute antiviral canonical pathways may markedly influence CHIKV disease severity.

Specifically, significant differences in the levels of several analytes were found in the virus-infected joints when comparing WT-CHIKV and RH-CHIKV infection, namely a decrease in GRO-α, and an increase in IL-27 and IL-10 in the RH-CHIKV-infected animals (Fig 3A). These differences may lead to changes in transcript expression levels and thus the associated canonical pathways, leading to suppressed inflammation. The anti-inflammatory effects of IL-10 have been previously associated with viral resolution during CHIKV infection in both patient cohorts and *in vivo* infections (Ng *et al*, 2009; Teng *et al*, 2015; Kulkarni *et al*, 2017). In addition, IL-10 has a protective immune modulatory role against neuro-adapted Sindbis virus-associated encephalomyelitis (Kulcsar *et al*, 2014). Although IL-27 has not been extensively studied in the context of alphavirus infection, IL-27 induces differentiation of diverse T-cell populations and up-regulates IL-10. GRO-α is a neutrophil chemoattractant that may increase infiltrating neutrophils into the inflamed joints, leading to a neutrophil-dominated response and exacerbated CHIKV disease pathology (Poo *et al*, 2014; Long & Heise, 2015). Concordantly, slightly lower levels of neutrophils were observed in the RH-CHIKV-infected group, which could have contributed to the decrease in disease pathology.

IPA of the immune mediators and publicly available transcriptome data also revealed associations of CHIKV infection with cell signaling canonical pathways involved in innate immunity activation and the transition to adaptive immunity (Fig 3E). Notably, MIP-1β (CCL4) and CSF2 transcripts (Fig 3F), which function as a macrophage chemoattractant and induce macrophage proliferation and activation, respectively, are increased in CHIKV-infected patients (Chow *et al*, 2011; Teng *et al*, 2015). Macrophages reportedly trigger the CHIKV-induced inflammatory cascade (Her *et al*, 2010); this feature is consistent with our findings, as RH-CHIKV infection exhibited decreased MIP-1β expression compared to WT-CHIKV, and slightly lower levels of macrophages (Fig EV3A), which could suppress joint inflammation. Furthermore, lower MIP-1β and CSF2 could have caused a decrease in activation of inflammatory monocytes during RH-CHIKV infection (Fig 4C–E), hence reducing pathology.

CD40 and IL-18 transcripts (Fig 3F) are also involved in activating adaptive immunity, and have been associated with CHIKV infection in patient cohort studies (Simarmata *et al*, 2016; Michlmayr *et al*, 2018). The co-stimulatory protein CD40 and pro-inflammatory cytokine IL-18 mediate T-cell activation, specifically IFN-γ secreting T helper 1 (Th1) cells (Dinarello, 1999). This feature is consistent with our findings, as RH-CHIKV showed decreased IL-18 and IFN-γ levels, indicative of a decreased Th1-cell activation and suppression of their pathogenic role in RH-CHIKV infection (Teo *et al*, 2013, 2017). Compared with WT-CHIKV, RH-CHIKV infection resulted in reduced IL-1β production (Fig 3D). IL-1β and IL-18 are up-regulated during the acute phase of CHIKV infection in both patients and *in vivo* models, acting in the NLRP3 inflammasome pathway and contributing to CHIKV pathology (Chen *et al*, 2017). Lower IL-1β and IL-18 levels in RH-CHIKV infection may lead to dampened NLRP3 pathways, resulting in reduced joint inflammation during acute phase of infection (Chen *et al*, 2017).

Consistent with the attenuated RH-CHIKV phenotype in this study, IL-1β and IFN-γ are important mediators for B-cell activation and maturation (Finkelman *et al*, 1988; Lowenthal *et al*, 1998; Nakae *et al*, 2001), and IL-1β is also a marker of CHIKV severity in patients (Ng *et al*, 2009). Despite the lower levels of these cytokines

in the RH-CHIKV-vaccinated group, there was no impact on the neutralizing capacity of the antibodies upon WT-CHIKV re-challenge compared to the control group (Fig EV5B). Furthermore, vaccination with the RH-mutant could still cross protect against the antigenically related ONNV (Partidos *et al*, 2012), highlighting its potential for the development as a pan-alphavirus vaccine. In conclusion, this study highlights the importance of the molecular processes governing CHIKV replication and pathogenesis, and the possible role of several immune mediators as drivers of pathology that do not affect the protective antibody response. These results indicate that the RH-CHIKV mutant produces a minimal CHIKV clinical phenotype upon exposure yet exhibits conserved immunogenicity with WT-CHIKV. These features render RH-CHIKV an interesting CHIKV vaccine candidate with associated cross-protection against other alphaviruses. In addition, the use of molecular manipulation methods to achieve live attenuation could be explored for cost-effective interventions against other viral-associated diseases.

## Materials and Methods

### Wild-type (WT) and mutant CHIKV infectious clones

The WT-CHIKV isolate (LR2006-OPY1) used for challenge experiments was isolated from an infected patient during the Reunion Island Outbreak in 2005–2006 (Bessaud *et al*, 2006a). The WT-ONNV isolate used for cross-protection challenge experiments was isolated from an infected patient from Chad in 2004 (Bessaud *et al*, 2006b). NsP-mutant CHIKV infectious clones were generated by recombination technology, PCR-based site-directed mutagenesis, and sub-cloning from the full-length infectious cDNA clone of the CHIKV LR2006-OPY1 isolate, as previously described (Saul *et al*, 2015). Mutant viruses were rescued in BHK-21 cells (Pohjala *et al*, 2011), and viruses were propagated in C6/36 (Kam *et al*, 2012b), as previously described. An infectious clone of this isolate containing ZsGreen under the control of a sub-genomic promoter (Lum *et al*, 2018) was used for neutralization assays.

### Production of virus stocks

Virus stocks used in *in vitro* experiments were propagated in Vero E6 cells, as previously described (Her *et al*, 2010). Virus stocks used in murine *in vivo* experiments were propagated in C6/36 *Aedes albopictus* cells (ATCC) and purified by sucrose-gradient ultracentrifugation, as previously described (Kam *et al*, 2012b). Viral stock titers were determined by standard plaque assay using Vero E6 cells (ATCC).

### *In vitro* mouse tail fibroblast (MTF) infection and infectivity quantification

Primary MTFs were isolated from the tails of male WT C57BL/6 mice aged 10 weeks, as previously described (Salmon, 2005). Virus infections were performed with the respective CHIKV nsP-mutants at a multiplicity of infection (MOI) of 10 in serum-free Dulbecco's modified Eagle medium (DMEM; HyClone). Virus overlay was removed after incubation for 1.5 h at 37°C and replenished with fresh DMEM supplemented with 10% fetal bovine serum (FBS;

HyClone). Mock infections were carried out in parallel. Infected MTFs were harvested 12 h post-infection (hpi) for infectivity quantification. CHIKV infection was directly quantified based on ZsGreen signal detection under the FITC channel (Lum *et al*, 2018), and data were acquired with a BD FACSCanto II flow cytometer (BD Biosciences). Viral RNA was extracted from 140 µl cell mixture using a QIAamp viral RNA kit (Qiagen). The viral load was determined by quantitative reverse transcription PCR (qRT–PCR) with a probe that detects the nsP1 encoding part of CHIKV genomic RNA (Plaskon *et al*, 2009).

### Pulse-chase analysis

BHK-21 cells were starved at 37°C for 30 min in methionine- and cysteine-free DMEM (Sigma-Aldrich) at 3 hpi. Cells were pulse-labeled with (50 mCi) $^{35}$S-labeled methionine and cysteine (Perkin Elmer) at 37°C for 15 min. The pulsed samples were chased for 45 min in medium containing an excess of unlabeled methionine and cysteine (37°C, 45 min).

Cells were washed with PBS and then lysed in 1% SDS. Proteins were subsequently denatured by boiling (100°C, 3 min). Samples were diluted 1:20 with NET buffer [50 mM Tris (pH 7.5), 150 mM NaCl, 5 mM EDTA, 0.5% NP-40] and incubated for 1 h at 4°C with combinations of rabbit polyclonal antisera against nsPs (generated in-house). Immunocomplexes were precipitated with protein A Sepharose CL-4B (Amersham Biosciences) overnight at 4°C. The precipitates were washed four times with NET buffer containing 400 mM NaCl. The precipitated proteins were denatured by heating in Laemmli buffer [100 mM Tris–HCl (pH 6.8), 4% SDS, 20% glycerol, 200 mM dithiothreitol (DTT), and 0.2% bromophenol blue], and the samples were separated by 8% SDS–PAGE. Images were acquired by exposure to Typhoon imager (GE Healthcare).

### Mice

Male wild-type (WT) and µMT C57BL/6J mice aged 3 weeks were bred and maintained under specific pathogen-free conditions at the Biological Resource Center (BRC) of the Agency for Science, Technology, and Research, Singapore (A*STAR). At least 5 mice were used per group for all experiments. Littermates were randomly assigned to experimental groups. All experimental procedures involving mice were approved by the Institutional Animal Care and Use Committee (IACUC 181353) of A*STAR, and in compliance with the guidelines of the Agri-Food and Veterinary Authority (AVA) and the National Advisory Committee for Laboratory Animal Research of Singapore (NACLAR).

### Virus infection and disease evaluation

Mice were inoculated subcutaneously in the ventral side of the right hind footpad with 1E6 plaque-forming units (PFU) of the respective virus suspended in 30 µl Dulbecco's phosphate-buffered saline (DPBS). Viremia levels were monitored daily from 1 day post-infection (dpi) until 8 dpi, and then every alternate day until 14 dpi. Joint swelling of the virus-inoculated foot was measured daily from 0 to 14 dpi, as previously described (Kam *et al*, 2012b). Height (thickness) and breadth measurements were made of the metatarsal region of the foot, and quantified as height × breadth. The disease score was expressed as the relative fold change in foot size compared with pre-infected foot (0 dpi), using the following formula: $[(x - 0 \text{ dpi})/0 \text{ dpi} \times 100]$, where $x$ is the quantified foot-pad measurement for each respective day.

### Viral RNA extraction and viral copies quantification

Blood (10 µl) was obtained from the tail vein and diluted in 120 µl DPBS supplemented with 10 µl citrate-phosphate-dextrose solution (Sigma-Aldrich). Purification of viral RNA from the blood samples was performed using a QIAamp Viral RNA Kit (Qiagen), according to the manufacturer's instructions. Viral load was quantified by qRT–PCR, as previously described (Plaskon *et al*, 2009). For ONNV viral genome quantification, the following primers were designed to amplify negative nsP1 viral RNA: forward primer (AATTACGCGA GAAAACTTGCG), reverse primer (TTTTTCCAGAGATGTTTTTATC TGT), and TaqMan probe (CCGCTGGAAAGGT). Similar amplification conditions were used for ONNV nsP1 primers as described in previous studies (Plaskon *et al*, 2009).

### Cytokine profiling in the joints

Mice were anesthetized by intraperitoneal injection of 150 mg of ketamine and 10 mg/kg xylazine cocktail, followed by intracardial perfusion with PBS at the indicated time-points. Subsequently, the footpads were collected and homogenized in 1.5 ml RIPA buffer (50 mM Tris–HCl pH7.4; 1% NP-40; 0.25% sodium deoxycholate; 150 mM NaCl; 1 mM EDTA) with 1× protease inhibitors (Roche) using a gentleMACS M tube with a gentleMACS Dissociator (Miltenyi). Cell lysates were then sonicated at 70% intensity for 15 s (Branson Ultrasonics Sonifier™ S-450), and the supernatants were collected for cytokine/chemokines quantification as described below. Data are expressed as pg/ml in footpad lysate.

### Multiplex microbead immunoassay for cytokine quantification

Cytokine and chemokine concentrations in mice serum and footpad lysates were quantified simultaneously using a multiplex microbead-based immunoassay, ProcartaPlex mouse Cytokine & Chemokine 36-plex Panel 1A (EPX360-26092-901; Thermo Scientific) following the manufacturer's protocol. Data were acquired using a Luminex FlexMap 3D® instrument (Millipore) and analyzed with Bio-Plex Manager™ 6.0 software (Bio-Rad) based on standard curves plotted through a five-parameter logistic curve setting. The cytokines and chemokines assayed included: IFN-γ, IL-12p70, IL-13, IL-1β, IL-2, IL-4, IL-5, IL-6, TNF-α, GM-CSF, IL-18, IL-10, IL-17A, IL-22, IL-23, IL-27, IL-9, GRO-α, IP-10, MCP-1, MCP-3, MIP-1α, MIP-1β, MIP-2, RANTES, eotaxin, IFN-α, IL-15/IL-15R, IL-28, IL-31, IL-1α, IL-3, G-CSF, LIF, ENA-78/CXCL5, and M-CSF. A two-plex microbead-based immunoassay using ProcartaPlex Mouse IFN alpha/IFN beta Panel (EPX020-22187-901; Thermo Scientific) was also performed to determine IFN-α and IFN-β levels in footpad lysates and in MTFs during CHIKV nsP-mutant infection.

### Bioinformatic analyses of luminex and transcriptomic data

A previous study performed a comprehensive RNA-sequencing analysis of CHIKV-infected joints during peak viremia (2 dpi) and acute

arthritis (7 dpi; Wilson et al, 2017). Here, transcriptome data were extracted. Analyte concentrations in joint cell lysates, comprising cytokines and chemokines, were quantified using a multiplex microbead-based immunoassay, ProcartaPlex mouse Cytokine & Chemokine 36-plex Panel 1A (EPX360-26092-901; Thermo Scientific) following the manufacturer's protocol, as described earlier. Both the extracted transcriptomic profiles and the quantified analytes were categorized based on their profiles across two critical phases: (i) at 2 days post-WT-CHIKV infection (2 dpi) relative to mock infection (0 dpi; background); and (ii) at peak joint swelling on 6 dpi (or 7 dpi for transcripts) relative to 2 dpi. Categorization was performed based on up-regulated (up), down-regulated (down) levels, or no difference (ND). This sequential categorization generated the analyte or DEG state: ND-ND; ND-up; ND-down; up-ND; up-up; up-down; down-ND; down-up; and down-down. Analytes in complete or partial agreement (same trend on at least one phase) with their corresponding transcripts were first identified and categorized by their states. The remaining transcripts with $Log_2$ fold change > 2 were then categorized by their states. These transcripts were subsequently used to discover canonical pathways along with the analytes in the same state using ingenuity pathway analysis (IPA). Involvement of up-down analytes with acute antiviral canonical pathways was established by expanding the IPA to include directly linked molecules (Dataset EV2). A comparison of individual analyte concentrations between WT and attenuated CHIKV infection was performed to identify the analytes that differ during infection.

## Mouse joint cell isolation

Mice were infected with the respective CHIKV nsP-mutants and euthanized at 6 dpi. The footpads and ankles were removed, deskinned, and placed immediately in 4 ml digestion medium containing dispase (2 U/ml; Invitrogen), collagenase IV (20 μg/ml; Sigma-Aldrich), and DNase I mix (50 μg/ml; Roche Applied Science) in complete Roswell Park Memorial Institute medium. The cells were then isolated as previously described (Teo et al, 2013).

## Phenotyping of leukocytes

Isolated joint cells were first blocked with 1% mouse/rat serum (Sigma-Aldrich) blocking buffer for 10 min. The cells were then stained for 20 min with the following Abs: BUV395-conjugated anti-mouse CD45 (clone 30-F11; BD Biosciences), Pacific Blue-conjugated anti-mouse CD4 (clone RM4-5; BioLegend), CF594-conjugated anti-mouse CD8 (clone 53-6.7; BD Biosciences), PE-Cy7-conjugated anti-mouse CD3 (clone 17A2; BioLegend), APC-Cy7-conjugated anti-mouse Ly6C (clone HK1.4; BioLegend), Alexa Flour 700-conjugated anti-mouse MHC-II (clone M5/114.15.2; BioLegend), PerCP-Cy5.5-conjugated anti-mouse LFA-1 (clone H155-78; BioLegend), BV650-conjugated anti-mouse CD11b (clone M1/70; BioLegend), BV605-conjugated anti-mouse CD11c (clone N418; BioLegend), CF594-conjugated anti-mouse Ly6G (clone 1A8; BD Biosciences), eFluor450-conjugated anti-mouse B220 (clone RA3-6B2; eBioscience), PE-conjugated anti-mouse MerTK (clone DS5MMER; eBioscience), APC-conjugated anti-mouse CD64 (clone X54-5/7.1; BioLegend), Biotin-conjugated anti-mouse NK1.1 (clone PK136; eBioscience), and BUV737 streptavidin (BD Biosciences). Samples

were acquired on a LSR II flow cytometer (BD Biosciences) with FACSDiva software and analyzed using FlowJo software.

## High-dimensional analysis of flow cytometry data

Each sample was pre-gated on live $CD45^+$ singlet events and then randomly down-sampled to the lowest event size observed among the samples ($n = 9,405$). Logicle transformation was performed using the estimateLogicle function of the flowCore package. The dimensionality reduction algorithm UMAP (umap-learn Python package v2.4.0) was then run using 1,000 epoch, 15 nearest neighbors, and otherwise default parameters (Becht et al, 2018; preprint: McInnes & Healy, 2018). For clustering, PhenoGraph (R package Rphenograph v0.99.1) was used with default parameters ($k = 30$). To compute the Jensen–Shannon divergence across pairs of samples, the UMAP embedding was binned into $40 \times 40$ bins into which one virtual observation (as an uninformative prior) was added. These counts were then log2-transformed, and the JS divergence was computed as previously described (Amir et al, 2013).

## Serum collection

Serum from individual mice was collected by retro-orbital bleeding at the indicated time-points. After leaving the blood to clot, the cells were spun down to remove cell contaminants from the resulting serum. Serum aliquots were either stored at −20°C for luminex assays or heat inactivated at 56°C for 30 min before ELISA and viral neutralization assays.

## ELISA

Antibody (Ab) titers were assessed by virion-based ELISA, as previously described (Kam et al, 2012a,b,c). CHIKV-coated (1E6 virions/well in 50 μl DPBS) polystyrene 96-well MaxiSorp plates (Nunc) were blocked with PBS containing 0.05% Tween-20 (PBST) and 5% w/v nonfat milk for 1.5 h at 37°C. Sera from normal and infected animals were heat inactivated and serially diluted in Ab diluent (0.05% PBST + 2.5% w/v nonfat milk). The diluted sera (100 μl) were then added to each well and incubated for 1 h at 37°C. HRP-conjugated goat anti-mouse IgG was used. Total IgG quantification assays were performed using pooled sera of individual animals diluted at 1:500. All HRP-conjugated Abs were purchased from Santa Cruz. ELISAs were developed using TMB substrate (Sigma-Aldrich) and terminated with Stop reagent (Sigma-Aldrich). Absorbance was measured at 450 nm.

## Neutralization assay

Ab neutralizing activity was tested using an immunofluorescence-based cell infection assay in HEK293T cells. The WT-CHIKV LR2006-OPY1 infectious clone expressing sub-genomic ZsGreen protein was incubated with heat-inactivated mouse sera diluted in complete media (1:625,000, 1:125,000, 1:25,000, 1:16,000, 1:8,000, 1:5,000, 1:4,000, 1:2,000, 1:1,000, 1:500, 1:250), for 1 h at 37°C with gentle rocking (160 rpm). Virus–Ab mixtures were added at MOI = 5 to HEK293T cells seeded in a 96-well plate ($3 \times 10^4$ cells/well) and incubated for 18 h. The cells were then harvested, fixed

**The paper explained**

**Problem**
CHIKV outbreaks are still prevalent, with significant increase in incidences in the last decade. Given the acute and chronic disabilities caused by CHIKV infection and the associated economic burden, effective preventive measures are urgently needed to halt CHIKV global expansion. However, there is currently no full-length live-attenuated vaccine against CHIKV.

**Results**
Using mouse *ex vivo* and *in vivo* infection models, we show that the mutation of nsPs at key positions caused prominent changes in viral replication and infectivity. Particularly, R532H mutation on nsP1 caused attenuation, with significant reduction in infectivity and pathogenesis in *in vivo* murine model. Further investigation revealed that attenuation could be due to changes in anti-inflammatory IL-10 and pro-inflammatory IL-1β and IL-18 levels during RH-CHIKV infection that modified acute antiviral and cell signaling canonical pathways. However, this CHIKV nsP-mutant was still immunogenic and able to induce a protective antibody response upon WT-CHIKV challenge.

**Impact**
This study highlights the effect mutations on key sites in the CHIKV non-structural proteins have on host innate inflammatory responses. Particularly, the use of molecular manipulation methods to achieve live attenuation could be relevant to a broad spectrum of viruses and can be explored for cost-effective interventions against other viral-associated diseases.

with 4% paraformaldehyde, and acquired using a MACSQuant Analyzer (Miltenyi Biotec). Infected cells expressing ZsGreen were quantified with FlowJo v10.0.7 software (FlowJo, LLC). The percentage of infectivity of the neutralization group relative to the virus infection group was calculated as follows: % Infectivity = 100 × (% infection from neutralization group/% infection from virus infection group).

### Vaccination and challenge timeline

Mice were vaccinated subcutaneously in the ventral side of the right hind footpad with 1E6 PFU of the respective attenuated CHIKV in 30 μl DPBS. These vaccinated mice were then challenged via the same route with 1E6 PFU WT-CHIKV 3 months later. Viremia and joint inflammation were monitored as described, for 2 weeks following challenge. Serum was collected at pre-challenge, 3, 6, and 14 dpi for serological studies.

### Histology

Mice were euthanized on 6 dpi after challenge experiments and perfused with 10% neutral-buffered formalin (NBF). Virus-inoculated joints were harvested and fixed in 10% NBF for 24 h at room temperature. The joints were then decalcified in 5% formic acid and trimmed to three parts at 5-mm intervals. Sectioned tissues were routinely processed, sliced into 5-μm-thick sections, and stained with hematoxylin and eosin (H&E). Tissues were viewed under an Olympus BX53 upright microscope (Olympus Life Science, Japan), and images were captured with an Olympus DP71 digital color camera using Olympus DP controller and DP manager software. Histopathological assessment was performed blind by histopathologists, using a scoring method in each individual animal based on the presence of edema, inflammation, muscle necrosis, tendonitis, and synovitis, if any. Severity of the grades was assigned to the following scale: 0—no finding; 1—minimal; 2—mild; 3—moderate; 4—marked; and 5—severe, as previously described (Morrison *et al*, 2011).

### Statistical analyses

Data are presented as the means ± SD unless specified otherwise. Differences between groups and controls were analyzed using the unpaired non-parametric Mann–Whitney $U$ statistical test unless specified otherwise. Statistical analyses were performed in GraphPad Prism 7.0a (GraphPad Software). *P*-values considered statistically significant are represented with * for $P < 0.05$, ** for $P < 0.01$, and *** for $P < 0.001$.

**Expanded View** for this article is available online.

### Acknowledgements

The authors would like to thank the SIgN Flow Cytometry core for assistance with cytometry analyses, Chloe Boehm from Tufts University for assistance with experiments, Esther Mok from SIgN immune monitoring group for support in luminex assay, and the SIgN mouse core for support in animal breeding. The authors also thank Insight Editing London for review and editing of the manuscript prior to submission. This study was funded by SIgN, Agency for Science, Technology and Research (A*STAR) core grant and the Biomedical Research Council, A*STAR. Yi-Hao Chan and Cheryl Yi-Pin Lee are supported by A*STAR Graduate Scholarships. The funders had no role in study design, data collection and analysis, decision to publish, or preparation of the manuscript. Flow cytometry and multiplex soluble protein assay platforms are part of the SIgN Immunomonitoring platform and supported by a BMRC IAF 311006 grant and BMRC transition funds #H16/99/b0/011.

### Author contributions

LFPN, AM, and F-ML conceived and supervised the study. Y-HC, AU, T-HT, JJLT, SNA, FAB, W-XY, CY-PL, and F-ML performed the experiments. RR performed blind histopathological scoring of the infected joints. Y-HC, T-HT, FAB, BL, EB, EN, RR, and F-ML performed data analysis. AM and AU constructed the viral infectious clones. Y-HC, T-HT, GC, F-ML, and LFPN wrote the manuscript.

### Conflict of interest
The authors declare that they have no conflict of interest.

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
