## [Review Process File · EMBO Molecular Medicine]

Mutating chikungunya virus non-structural protein produces potent live-attenuated vaccine candidate

Yi-Hao Chan, Teck-Hui Teo, Age Utt, Jeslin J. L. Tan, Siti Naqiah Amrun, Farhana Abu Bakar, Wearn-Xin Yee, Etienne Becht, Cheryl Yi-Pin Lee, Bernett Lee, Ravisankar Rajarethinam, Evan Newell, Andres Merits, Guillaume Carissimo, Fok-Moon Lum, Lisa F.P. Ng

Review timeline:	Submission date:	20 November 2018
	Editorial Decision:	16 January 2019
	Revision received:	9 March 2019
	Editorial Decision:	25 March 2019
	Revision received:	27 March 2019
	Accepted:	28 March 2019

Editor: Céline Carret

Transaction Report:

1st Editorial Decision

16 January 2019

Thank you for the submission of your manuscript to EMBO Molecular Medicine and please accept my sincere apologies for the delay in obtaining the referees' reports. We have now heard back from two referees.

You will see that both reports are supportive of publication and both referees have important and overlapping suggestions that we believe would improve the translational aspect of the paper. The most important aspect of this study that needs addressing concerns vaccine stability and safety, and referees recommend performing a safety/mutational study to see how stable is the mutation R532H in vivo or at least in vitro.

We would therefore welcome the submission of a revised version of this article within three months for further consideration and would like to encourage you to address all the criticisms raised as suggested to improve conclusiveness and clarity. Please note that EMBO Molecular Medicine strongly supports a single round of revision and that, as acceptance or rejection of the manuscript will depend on another round of review, your responses should be as complete as possible.

I look forward to receiving your revised manuscript.

***** Reviewer's comments *****

Referee #1 (Remarks for Author):

Chan et al described the potential use of point mutants of chikungunya virus as attenuated vaccines. Amongst these mutants, the one with R532H mutation at the NSP1 has very promising data. Mice vaccinated with this mutant have low virus replication and very mild pathology. This group vaccinated mice can be protected from a WT virus challenge. In addition, the protection can cross react with a closely related alphavirus, O'nyong-nyong virus.

This is an excellent study. The work is very comprehensive. All the in vitro and vivo studies were nicely conducted and results were well presented. The primary mechanism of the protection (i.e. neutralizing antibody) was clearly demonstrated. This reviewer has enjoyed reading this manuscript. With the exceptions of a few very minor comments, this reviewer has no major concern on this work.

Minor comments:

1. This reviewer agrees that the R532H mutant has vaccine potential. But the reviewer has some concern about its genetic stability. An attenuated vaccine with only one mutation might have a safety concern. Can the authors comment on this? Would the authors recommend introducing other mutations to prevent its reversion? Other alternative strategies are welcomed.
2. Acute antiviral responses might be beneficial in some conditions, but an over-reacting one might also cause some harmful effects (e.g. cytokine dysregulation observed in severe viral pneumonia). Can the authors comment on the magnitudes of cytokine/chemokine productions induced by the RH and RHEV mutants? It is better to highlight that these responses are not extreme and, more importantly, are well-regulated ones.
3. Describe how the WT chikungunya viral proteins can suppress innate sensing.
4. Any possible explanations of muscle regeneration caused by the WT, but not the RH mutant?
- 5.(Optional) Did the authors test their mutants in Vero cells that have a genetic defect in interferon production?

Referee #2 (Comments on Novelty/Model System for Author):

No ethical issues are raised. For specific comments see below.

Referee #2 (Remarks for Author):

Mutating Chikungunya virus non-structural protein produces potent live-attenuated vaccine candidate

In the current manuscript the authors generate three mutant CHIKV strains containing mutations in nsP1 (R532H - RH) and/or nsP2 (E515V - EV) that have different levels of attenuation. The goal of the manuscript is to determine whether these viruses would make efficacious and safe vaccines against CHIKV challenge. The RH and RHEV mutants were severely attenuated in mice and displayed reduced joint inflammation but had the highest levels of Type 1 IFN induction at 15 hpi. All of the constructs were capable of eliciting antibody responses and protected mice following subsequent challenge with CHIKV or the related alphavirus ONNV. The authors performed extensive cytokine and immune cell profiling allowing them to make suggestions about how the attenuated strains protect against joint disease.

Overall the manuscript is well written and the results are interesting and may prove to be important if this approach were used to vaccinate people against CHIKV or applied to other alphaviruses. The clarity of the manuscript would be improved from the following:

It is unclear why in Figure 1C that the mutant EV has higher viremia and disease than WT virus. The levels are quite substantial. The authors should consider sequencing the virus from this animal experiment to determine whether it still contains the desired mutation. These viruses mutate rapidly. In fact, a safety/mutational study should be performed to determine how many passages it takes the virus to revert to a WT geno/phenotype and whether this happens in vivo.

The data shown in figure EV3 is confusing and should be omitted. Data shown in Figure EV3B-G should be validated by qRT-PCR to prove that CD4 and CD8 T cells contain virus and that it is not an artifact. This experiment needs to be carefully done and the authors should quantify the percentage of virus present in leukocyte vs. non-leukocyte populations.

Supplemental Table 2 is in disrepair and should be modified to improve readership quality.

Line 166/Figure 2. It doesn't appear that IL-15 and GM-CSF are increased in RH or RHEV as suggested in the text. This should be addressed. Similarly on Line 171 it is suggested that IL4 and IL5 were upregulated in the mutants at 15 h but this doesn't appear to be the case from the data presented in Figure 2.

Should Line 198 be limited to Figure 3B and not both 3A and 3B?

Line 207 would benefit by defining the time point for Gro-a, IL27 and IL10 differences.

While the phenotypic analysis of joint cells is fascinating there are a couple of issues that arise with Figure 4 that should be addressed. First, it appears that most of the phenographic clusters are robustly positive for CD11c and CD4. Is this correct since it would suggest that they all contain macrophages or dendritic cells? If so it needs to be explained. Also why are the CD8+ cells within clusters 15, 5, 16, 27 and 21 negative for CD3? And in addition most of the CD4+ cells are low for CD3 as well. These discrepancies need to be addressed.

The statistical analysis in Figure 5B is odd along with the description of neutralization efficiency in Line 260 should be addressed. Figure 5B. IC90/IC50 values for neutralization would be easier to interpret and compare between samples.

1st Revision - authors' response

9 March 2019

Point-by-point response to reviewers' comments EMM Submission EMM-2018-10092

Reviewer #1

(General comments for Author):

Chan et al describe the potential use of point mutants of chikungunya virus as attenuated vaccines. Amongst these mutants, the one with R532H mutation at the nsP1 has very promising data. Mice vaccinated with this mutant have low virus replication and very mild pathology. This group vaccinated mice can be protected from a WT virus challenge. In addition, the protection can cross react with a closely related alphavirus, O'nyong-nyong virus.

This is an excellent study. The work is very comprehensive. All the in vitro and in vivo studies were nicely conducted and results were well presented. The primary mechanism of the protection (i.e. neutralizing antibody) was clearly demonstrated. This reviewer has enjoyed reading the manuscript. With the exceptions of a few very minor comments, this reviewer has no major concern on this work.

Response: We thank the reviewer for the encouraging comments.

Minor comments:

1. This reviewer agrees that the R532H mutant has vaccine potential. But the reviewer has some concern about its genetic stability. An attenuated vaccine with only one mutation might have a

safety concern. Can the authors comment on this? Would the authors recommend introducing other mutations to prevent its reversion? Other alternative strategies are welcomed.

Response: We thank the reviewer for this comment. It is a valid concern to know the genetic stability of the R532H mutant. As recommended, we have conducted an *in vitro* serial passage of R532H in VeroE6 cells from the infection stock up to passage 5, and harvested the virus at each passage based on the cytopathic effect shown (50%). Each passage took 30 to 40 hours before 50% CPE was observed. The region surrounding R532H mutation (in red) was sequenced, and the mutation remained stable. The results are shown below for review purposes.

G A C A G A G C G
 C A C

(Magnified view)

2. Acute antiviral responses might be beneficial in some conditions, but an over reacting one might also cause some harmful effects (e.g. cytokine dysregulation observed in severe viral pneumonia). Can the authors comment on the magnitudes of cytokine/chemokine productions induced by the RH and RHEV mutants? It is better to highlight that these responses are not extreme and, more importantly, are well-regulated ones.

Response: We thank the reviewer for raising this concern. The increase in cytokine and chemokine production during RH-CHIKV and RHEV-CHIKV are not extreme and well regulated. An example will be early IFN- γ response in the infected joints (15 hpi), where RH-CHIKV and RHEV-CHIKV infected mice produce 5 times more IFN- γ than WT-CHIKV infection. However, this up-regulation in IFN- γ is well-regulated and difference in response between infected groups are neutralized at 2 dpi. We have now revised the manuscript on page 9, line 180 to 183 to read “The well-regulated immune responses in the RH-CHIKV and RHEV-CHIKV infected mice indicate that an early increase in pro-inflammatory mediators (15 hpi) followed by gradual depletion from 2 to 6 dpi is associated with suppressed disease pathology.”

3. Describe how the WT chikungunya viral proteins can suppress innate sensing.

Response: We thank the reviewer for the comment to improve the manuscript. We have now revised the manuscript on page 5, line 82 to 85 to read “Another study showed that CHIKV nsP2 inhibited type-I IFN signaling and reduced expression of anti-viral mediators in the host by inhibiting IFN stimulated JAK/STAT signaling and suppressing IFN-induced gene expression (Fros et al., 2010).”

4. Any possible explanations of muscle regeneration caused by the WT, but not the RH mutant?

Response: The muscle regeneration seen in WT-CHIKV mice in Figure 6 could be due to the higher extent of muscle damage in the mouse during the vaccination with WT-CHIKV compared to RH-CHIKV. The explanation to this question has been included in the revised manuscript on page 14, lines 303 to 304 to read “The difference in regenerated muscle fibers could be due to higher extent of muscle damage in mice vaccinated with WT-CHIKV.”

5. (Optional) Did the authors test their mutants in vero cells that have a genetic defect in interferon production?

Response: We did not test the mutants in VeroE6 cells that have a genetic defect in interferon production as this is not the best cell line to screen for interferon production.

Reviewer #2

(General comments for Author):

Mutating Chikungunya virus non-structural protein produces potent live-attenuated vaccine candidate

In the current manuscript, the authors generate three mutant CHIKV strains containing mutations in nsP1 (R532H - RH) and/or nsP2 (E515V - EV) that have different levels of attenuation. The goal of the manuscript is to determine whether these viruses would make efficacious and safe vaccines against CHIKV challenge. The RH and RHEV mutants were severely attenuated in mice and displayed reduced joint inflammation but had the highest levels of Type 1 IFN induction at 15 hpi. All of the constructs were capable of eliciting antibody responses and protected mice following subsequent challenge with CHIKV or the related alphavirus ONNV. The authors performed extensive cytokine and immune cell profiling allowing them to make suggestions about how the attenuated strains protect against joint disease.

Overall the manuscript is well written and the results are interesting and may prove to be important if this approach were used to vaccinate people against CHIKV or applied to other alphaviruses. The clarity of the manuscript would be improved from the following:

Response: We thank the reviewer for the encouraging comments.

1. It is unclear why in Figure 1C that the mutant EV has higher viremia and disease than WT virus. The levels are quite substantial. The authors should consider sequencing the virus from this

animal experiment to determine whether it still contains the desired mutation. These viruses mutate rapidly. In fact, a safety/mutational study should be performed to determine how many passages it takes the virus to revert to a WT geno/phenotype and whether this happens *in vivo*.

Response: An *in vitro* mutational study was performed for the vaccine candidate (RH-CHIKV) and reported in a reply to reviewer 1's first comment. We also went on to investigate if reversion to WT-CHIKV happened *in vivo*, by isolating and sequencing viral RNA in RH-CHIKV infected joint cell lysate at 2 dpi (peak of viremia). Results also showed that the mutation was stable. Region surrounding R532H mutation (in red) was sequenced, and illustrated below for review purposes.

2. The data shown in figure EV3 is confusing and should be omitted. Data shown in Figure EV3B-G should be validated by qRT-PCR to prove that CD4 and CD8 T cells contain virus and that it is not an artifact. This experiment needs to be carefully done and the authors should quantify the percentage of virus present in leukocyte vs. non-leukocyte populations.

Response: We thank the reviewer for the comment and we have removed figure EV3 as recommended. Only percentage infection detected by FACs in leukocyte vs non-leukocyte populations will remain to show the drop of infectivity by the CHIKV nsP mutants. We have now revised the manuscript on Page 7 and 8, line 145 to 148 to read “All three CHIKV nsP-mutants showed significantly reduced virus infection in joint footpad CD45+ leukocytes at 2 dpi (Fig 1E). In addition, the reduced virus infectivity of all CHIKV nsP-mutants was also observed in CD45-negative cells in the CHIKV-infected joints (Fig EV3B).” The edited figure is as attached below for easy reference

Figure EV3. R532H and E515V mutations in CHIKV nsPs result in lower infectivity at the site of inflammation.

WT C57BL/6 mice were infected subcutaneously with 1E6 PFU ZsGreen-tagged WT CHIKV, RH CHIKV, EV CHIKV and RHEV CHIKV at the metatarsal region of the footpad.

(A) Representative fluorescence activated cell sorting gating strategy to isolate specific leukocyte subsets.

(B) Infection at both 2 dpi and 6 dpi were assessed in CD45- cells. Background ZsGreen signals are indicated by the dotted line. The data are presented as the means \pm SD (n=5 per group). Statistical analyses were performed using two-tailed Mann Whitney U test (** P = 0.008 RH-CHIKV 2 and 6 dpi, ** P = 0.008 RHEV-CHIKV 2 and 6 dpi, ** P = 0.008 EV-CHIKV 2 and 6 dpi).

Figure EV3 Chan *et al.*, 2019

3. Supplemental Table 2 is in disrepair and should be modified to improve readership quality.

Response: Supplemental Table 2 has been submitted in excel format to improve readership quality.

4. Line 166/Figure 2. It doesn't appear that IL-15 and GM-CSF are increased in RH or RHEV as suggested in the text. This should be addressed. Similarly on Line 171 it is suggested that IL4 and

IL5 were upregulated in the mutants at 15 h but this doesn't appear to be the case from the data presented in Figure 2.

Response: We have now revised the manuscript on page 8, line 164 to 168 and page 9, line 169 to 170. The sentences have been edited to “At 15 hpi, pro-inflammatory mediators (IL-1b, TNF-a, IL-15), chemokines (GRO-a, MCP-1, RANTES, MIP-1b, MIP-1a) and hematopoietic growth factors (GM-CSF) were significantly higher in the virus-infected joints of RH-CHIKV and RHEV-CHIKV mutants relative to WT-CHIKV (Fig 2, Table EV1). These findings are consistent with the high levels of IFN-a and IFN-b in these mutants (Fig 1F). Higher levels of anti-inflammatory cytokines IL-4 and IL-5 were also observed in these mutants compared to WT-CHIKV (Fig 2, Table EV1).”

5. Should Line 198 be limited to Figure 3B and not both 3A and 3B?

Response: We have now revised the manuscript on page 9, line 200 to read “Four of the top five canonical pathways found were involved in acute antiviral responses (Fig 3B).”

6. Line 207 would benefit by defining the time point for Gro-a, IL27 and IL10 differences.

Response: We have now amended the text on page 10, line 205 to read “Comparing all CHIKV nsP-mutants with WT-CHIKV infected joints revealed differences in GRO-a at 2 dpi, and IL-27 and IL-10 at 6 dpi in RH-CHIKV infected joints, which participate in these acute antiviral canonical pathways (Fig 3A).”

7. While the phenotypic analysis of joint cells is fascinating there are a couple of issues that arise with Figure 4 that should be addressed. First, it appears that most of the phenographic clusters are robustly positive for CD11c and CD4. Is this correct since it would suggest that they all contain macrophages or dendritic cells? If so it needs to be explained. Also why are the CD8+ cells within clusters 15, 5, 16, 27 and 21 negative for CD3? And in addition most of the CD4+ cells are low for CD3 as well. These discrepancies need to be addressed.

Response: We wish to explain that it is important to note that the B220 and CD4 specific antibody (same for CD8 and Ly6G) used for phenotyping of leukocytes have the same fluorophore as stated in the materials and methods (Pages 24 and 25). Therefore, clusters from Figure 4 must be identified based on the collective expression of lineage markers, i.e. the co-expression of CD3 with CD4 or CD8 to identify CD4+ or CD8+ T cells respectively, or Ly6G with CD11b for neutrophils. To improve clarity, we have now revised the manuscript on page 11, lines 231 to 232 to read “Cluster identity should be based on the collective expression levels of lineage surface markers.”

8. The statistical analysis in Figure 5B is odd along with the description of neutralization efficiency in Line 260 should be addressed. Figure 5B. IC90/IC50 values for neutralization would be easier to interpret and compare between samples.

Response: We agree with the reviewer’s suggestion and performed neutralization assays to find out the IC50 of sera collected prior to WT-CHIKV challenge. We have now revised the manuscript on page 12, lines 262 to 273 to read “Concordantly, antibodies generated by RH-CHIKV and RHEV-CHIKV-vaccinated mice were able to neutralize CHIKV infection from 6 dpv (Fig EV6A). To qualify as a protective CHIKV vaccine, the vaccine candidates should induce lasting CHIKV neutralizing antibodies. Indeed, all three CHIKV nsP-mutant vaccinated groups showed similar titers of CHIKV-specific IgG at 90 dpv (Fig 5A). Half maximal inhibitory concentration (IC₅₀) of the sera collected prior to WT-CHIKV challenge at 90 dpv was also investigated. Notably, these antibodies retained their neutralizing capacity (Fig 5B). Although the neutralizing capacity of antibodies for RH-CHIKV and RHEV-CHIKV were lower than WT-CHIKV vaccinated mice at 90 dpv, shown by the lower dilution levels to reach IC₅₀ (RH-CHIKV (1:1388) < RHEV-CHIKV (1:1807) < WT-CHIKV (1:3819), the levels of CHIKV neutralizing antibodies induced are sufficient for protection *in vivo* when the vaccinated mice were challenged with WT-CHIKV (Fig 5C).”

The revised Figure 5 and Figure EV6 are illustrated below for easy reference.

Figure 5. Vaccination with CHIKV nsP-mutants induces neutralizing antibodies and protects against pathology during CHIKV challenge. (A) Anti-CHIKV IgG titers from the vaccination phase were determined with a CHIKV virion-based ELISA. Sera from four animals were pooled and diluted 1:500 for the assay. Statistical analyses were performed using two-tailed paired T-test ($*P = 0.029$ RH-CHIKV 6 dpv, $*P = 0.029$ RH-CHIKV 90 dpv, $*P = 0.029$ RHEV-CHIKV 6 dpv, $*P = 0.029$ RHEV-CHIKV 90 dpv, $*P = 0.029$ EV-CHIKV 90 dpv). (B) Half maximum inhibitory concentration (IC_{50}) of pooled sera collected on 90 dpv were characterized in HEK293T cells infected with WT-CHIKV (MOI 5). Sera were serially diluted at 1:625000, 1:125000, 1:25000, 1:16000, 1:8000, 1:5000, 1:4000, 1:2000, 1:1000, 1:500, 1:250. Sera dilution to obtain IC_{50} are shown in parenthesis. Data shown are representative of three independent experiments. (C) WT C57BL/6 mice were vaccinated subcutaneously with $1E6$ PFU of WT-CHIKV or CHIKV nsP-mutants (RH, RHEV and EV) at the metatarsal region of the footpad. Challenge with $1E6$ PFU of WT-CHIKV was performed via the same route at 90 dpv. (D,E) (D) Viremia and (E) severity of joint inflammation were monitored over 2 weeks. Viremia is detected with CHIKV nsP1 probe via qRT-PCR. The data are presented as the means \pm SD and are representative of two independent experiments ($n=6$ per group).

Figure 5 Chan et al., 2019

Figure EV6. Vaccination with attenuated CHIKV induces a robust antibody response against WT-CHIKV challenge.

(A) The neutralizing capacity of the pooled sera was characterized at 1:1000 dilution in HEK293T cells infected with WT-CHIKV (MOI 5). The percentage infection was normalized to sera isolated from mock-vaccinated mice. Tested sera were obtained at 0, 3, 6, 14 and 90 dpv. Statistical analyses were performed using two-tailed Mann Whitney *U* test against the positive control (WT-CHIKV) on the respective dpv ($*P = 0.029$ RH-CHIKV 14 and 90 dpv, $*P = 0.029$ RHEV-CHIKV 90 dpv, $*P = 0.029$ EV-CHIKV 6, 14 and 90 dpv).

(B) Anti-CHIKV IgG titers were determined with a CHIKV virion-based ELISA. Sera from four animals were pooled and diluted 1:500 for the assay. Statistical analyses were performed using two-tailed Mann Whitney *U* test against mock-vaccinated group on each respective dpc ($*P = 0.029$ RH-CHIKV 3, 6 and 14 dpc, $*P = 0.029$ RHEV-CHIKV 3, 6 and 14 dpc, $*P = 0.029$ EV-CHIKV 3, 6 and 14 dpc).

(C) Neutralizing capacity of the pooled sera was characterized at a 1:1000 dilution in HEK293T cells infected with WT-CHIKV (MOI 5). Data are presented as the means \pm SD. The percentage infection was normalized to virus-only infection. Sera tested were obtained at 3, 6 and 14 dpc. Statistical analyses were performed using two-tailed Mann Whitney *U* test against mock-vaccinated group on each respective dpc ($*P = 0.029$ RH-CHIKV 3, 6 and 14 dpc, $*P = 0.029$ RHEV-CHIKV 3 and 6 dpc, $*P = 0.029$ EV-CHIKV 3, 6 and 14 dpc).

Figure EV6 Chan et al., 2019

Thank you for the submission of your revised manuscript to EMBO Molecular Medicine. We have now received the enclosed reports from the referee who was asked to re-assess it. As you will see the reviewer is now supportive and I am pleased to inform you that we will be able to accept your manuscript pending the minor editorial amendments.

***** Reviewer's comments *****

Referee #1 (Comments on Novelty/Model System for Author):

The authors have addressed all the concerns. I have no further comments.

Referee #1 (Remarks for Author):

Thanks for address the comments.

2nd Revision - authors' response

27 March 2019

Authors made the requested editorial changes.

Corresponding Author Name: Lisa F.P. Ng
Journal Submitted to: EMBO Molecular Medicine
Manuscript Number: EMM-2018-10092